# Structural basis for terminal loop recognition and stimulation of pri-miRNA-18a processing by hnRNP A1

Hamed Kooshapur [1,2], Nila Roy Choudhury[3], Bernd Simon[4], Max Mühlbauer [2,5], Alexander Jussupow[2,5], Noemi Fernandez[6], Alisha N. Jones[1,2], Andre Dallmann[1,2], Frank Gabel[7,8], Carlo Camilloni[2,5], Gracjan Michlewski [3,6,9,10], Javier F. Caceres [6] & Michael Sattler [1,2]

Post-transcriptional mechanisms play a predominant role in the control of microRNA (miRNA) production. Recognition of the terminal loop of precursor miRNAs by RNA-binding proteins (RBPs) influences their processing; however, the mechanistic basis for how levels of individual or subsets of miRNAs are regulated is mostly unexplored. We previously showed that hnRNP A1, an RBP implicated in many aspects of RNA processing, acts as an auxiliary factor that promotes the Microprocessor-mediated processing of pri-mir-18a. Here, by using an integrative structural biology approach, we show that hnRNP A1 forms a 1:1 complex with pri-mir-18a where both RNA recognition motifs (RRMs) bind to cognate RNA sequence motifs in the terminal loop of pri-mir-18a. Terminal loop binding induces an allosteric destabilization of base-pairing in the pri-mir-18a stem that promotes its downstream processing. Our results highlight terminal loop RNA recognition by RBPs as a potential general principle of miRNA biogenesis and regulation.

[1] Institute of Structural Biology, Helmholtz Zentrum München, Neuherberg 85764, Germany. [2] Center for Integrated Protein Science Munich at Department Chemie, Technical University of Munich, Garching 85747, Germany. [3] Wellcome Trust Centre for Cell Biology, University of Edinburgh, Michael Swann Building, Edinburgh EH9 3BF, UK. [4] Structural and Computational Biology Unit, European Molecular Biology Laboratory, Heidelberg 69117, Germany. [5] Institute for Advanced Study, Technical University of Munich, Garching 85747, Germany. [6] MRC Human Genetics Unit, Institute of Genetics and Molecular Medicine, University of Edinburgh, Western General Hospital, Edinburgh EH4 2XU, UK. [7] Institut Laue-Langevin, Grenoble 38044, France. [8] Univ. Grenoble Alpes, CEA, CNRS, IBS, Grenoble 38042, France. [9] Division of Infection and Pathway Medicine, University of Edinburgh, The Chancellor's Building, 49 Little France Crescent, Edinburgh EH16 4SB, UK. [10] ZJU-UoE Institute, Zhejiang University, 718 East Haizhou Road, Haining, Zhejiang 314400, China. Correspondence and requests for materials should be addressed to G.M. (email: Gracjan.Michlewski@ed.ac.uk) or to J.F.C. (email: Javier.Caceres@igmm.ed.ac.uk) or to M.S. (email: sattler@helmholtz-muenchen.de)

MicroRNAs (miRNAs, miRs) are a class of highly conserved small non-coding RNAs that play a crucial role in the regulation of gene expression. They are involved in a variety of biological processes including cell growth, proliferation, and differentiation[1]. Mature miRNAs are generated by two RNA cleavage steps involving nuclear and cytoplasmic RNase III enzymes (Drosha and Dicer, respectively). Primary miRNA (pri-miRNA, pri-mir) transcripts are cropped by the Microprocessor complex (comprising Drosha and DGCR8) in the nucleus giving rise to ~70 nucleotide (nt) stem-loop precursor miRNAs (pre-miRNAs, pre-mir), which following export to the cytoplasm, are further processed by Dicer into mature miRNAs[2]. In higher organisms, many miRNA genes are transcribed together as a cluster[3]. A prototypical example is the miR-17–92 cluster that is encoded as an intronic polycistron comprising six individual miRNAs that are highly conserved in vertebrates (miR-17, miR-18a, miR-19a, miR-20a, miR-19b-1, miR-92a-1)[4,5]. This cluster is frequently amplified and overexpressed in human cancers; hence, it is also referred to as oncomiR-1.

The biogenesis of miRNAs is tightly regulated and results in tissue-specific and developmental-specific expression patterns of miRNAs[6]. A number of specific RNA-binding proteins (RBPs) have recently emerged as important post-transcriptional regulators of miRNA processing. However, very little is known about their mechanism of action. Previously, we identified heterogenous nuclear ribonucleoprotein A1 (hnRNP A1) as an auxiliary factor that positively regulates the processing of miRNA-18a primary transcript (pri-mir-18a) by making specific contacts to the terminal loop of the RNA[7,8] (Fig. 1a, b). HnRNP A1 is a highly abundant RBP that has been implicated in diverse cellular functions related to RNA processing, including alternative splicing regulation[9,10], mRNA export[11], IRES (internal ribosome entry site)-mediated translation[12], mRNA stability[13], and telomere maintenance[14,15].

Here, we have combined an integrative structural biology approach with biochemical and functional assays to provide mechanistic insights into the role of hnRNP A1 in stimulating pri-mir-18a processing. We show that hnRNP A1 forms a 1:1 complex with pri-mir-18a in solution, with the recognition of two UAG motifs by the tandem RRM domains of hnRNP A1 revealed by a high-resolution crystal structure. NMR and biophysical data show that high-affinity binding involves recognition of two UAG motifs in the pri-mir-18a terminal loop and the proximal stem region. Notably, binding to the terminal loop induces an allosteric destabilization of base-pairing in the pri-mir-18a stem that promote its processing. These findings may serve as a paradigm for the regulation of miRNA processing by the recognition of the terminal loop by RBPs.

## Results

**UP1 stimulates pri-mir-18a processing**. We have previously shown that hnRNP A1 acts as an auxiliary factor that promotes miR-18a biogenesis; however, the underlying molecular mechanisms are unknown[7]. HnRNP A1 has two RNA recognition motif (RRM) domains, each harboring conserved RNP-1 and RNP-2 submotifs, and a C-terminal flexible glycine-rich tail, which includes the M9 sequence responsible for nuclear import and export[11,16]. The RNA-binding region of hnRNP A1, comprising the tandem RRM1-RRM2 domains, is referred to as UP1 (Unwinding Protein 1)[17] (Fig. 1a). To identify which regions in hnRNP A1 are required for stimulating pri-mir-18a processing in living cells we used an in vivo processing assay. For this, several N-terminal T7-tagged hnRNP A1 constructs were transiently overexpressed in HeLa cells and the level of mature miR-18a was analyzed by qRT-PCR. We found that overexpression of full-

length hnRNP A1 results in a ~two-fold increase in the levels of mature miR-18a, whereas UP1, comprising both RRMs but lacking the M9 sequence, has no effect—due to its cytoplasmic localization (Fig. 1c; Supplementary Fig. 1a, b; Supplementary Table 1). This was confirmed by transient expression of UP1-M9, where UP1 is fused to the M9 sequence that directs nuclear localization of hnRNP A1[16]. UP1-M9 localized exclusively to the nucleus (Supplementary Fig. 1b; Supplementary Table 1) and, importantly, displayed similar activity as full-length hnRNP A1 in stimulating miR-18a production in vivo (Fig. 1c). By contrast, RRM1-M9 and RRM2-M9 that partially localize to the nucleus (Supplementary Fig. 1b; Supplementary Table 1) did not increase miR-18a levels, showing that nuclear localized UP1 is necessary and sufficient to enhance pri-mir-18a processing (Fig. 1c). To assess the effect of RNA-binding in the functional activity of UP1, we mutated conserved Phe residues within RNP motifs of the two RRM domains that directly contact the RNA and are required for the RNA-binding activity in hnRNP A1[17]. Substitution of Phe with Asp within RRM2 (UP1-M9_FD2) or with Asp or Ala in both RRM1 and RRM2 domains (UP1-M9_FD12, UP1-M9_FD12a, or UP1-M9_FA12b) is sufficient to abolish the activity of UP1-M9 in our in vivo pri-mir-18a processing assay without affecting the nuclear localization of the protein constructs (Supplementary Fig. 1a, b; Supplementary Table 1). This indicates that the RNA-binding activity of hnRNP A1 is essential for enhancing miR-18a biogenesis. The in vivo functional data further confirm that the stimulatory function of hnRNP A1 in processing of pri-mir-18a requires both RRM domains, as mutations that affect RNA binding in one domain (or deletion of one domain) abolish the activity of hnRNP A1.

**UP1 specifically recognizes the loop region of pri-mir-18a**. To determine the regions of pri-mir-18a that are recognized by hnRNP A1 we performed electro-mobility shift assays (EMSA) with RNA variants corresponding to the terminal loop and stem of pri-mir-18a (Fig. 1b) and UP1, which has been shown to recapitulate most of the functions of full-length hnRNP A1 in vitro[17] (Fig. 1d). We observed that UP1 specifically binds to the terminal loop RNA, whereas no binding to the stem RNA was detected even at higher protein-to-RNA ratios. Single RRM1 and RRM2 domains do not show any detectable RNA-binding activity in this assay (Supplementary Fig. 1c). Altogether, these data show that (i) UP1 binds specifically to the terminal loop region of pri-mir-18a, and (ii) both RRM domains of UP1 are required for high-affinity RNA binding, indicating that they bind cooperatively.

**Minimal UP1-binding element in the pri-mir-18a terminal loop**. To provide a quantitative analysis of binding affinities of UP1 with pri-mir-18a, we performed isothermal titration calorimetry (ITC) experiments. Pri-mir-18a 71-mer RNA binds to UP1 with a dissociation constant ($K_D$) of 147 nM (Fig. 2a, right panel). To identify a minimal RNA region required for efficient UP1 binding, we tested RNA fragments of various sizes for binding to UP1 and individual RRMs by ITC (Table 1; Fig. 2a; Supplementary Fig. 2a). A 7-mer oligonucleotide (5′ AG<u>UAG</u>AU 3′) corresponding to the terminal loop of pri-mir-18a harbors an UAG motif, which is known to be recognized by hnRNP A1[18]. This 7-mer single-stranded RNA binds to the individual RRM1 and RRM2 domains with low micromolar affinity ($K_D = 20.4$ μM and 6.9 μM, respectively) forming 1:1 complexes (Table 1; Supplementary Fig. 2a). Binding of the 7-mer to UP1 has a $K_D$ of 3.4 μM (Table 1; Fig. 2a, left panel), suggesting that each RRM domain in UP1 can recognize the 7-mer RNA. Notably, a single-stranded 12-mer oligonucleotide (5′ AG<u>UAG</u>AU<u>UAG</u>CA 3′)

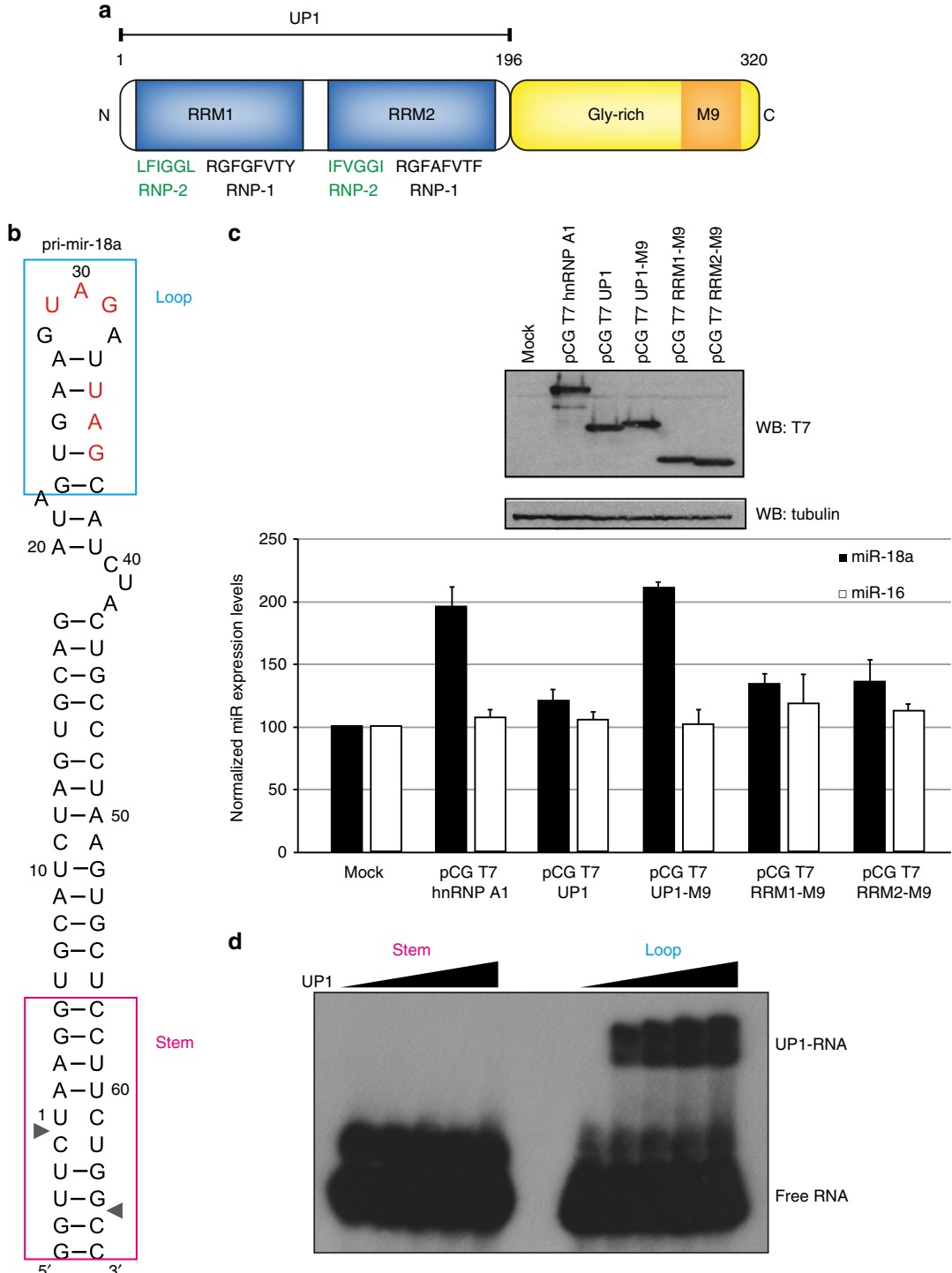

**Fig. 1** The tandem RRMs of hnRNP A1 promote pri-mir-18a biogenesis in living cells. **a** Domain structure of human hnRNP A1. The sequences of conserved RNP-1 and RNP-2 motifs in the RRM domains are indicated. M9 is a transport signal linked with both nuclear import and export of this protein. **b** Secondary structure of pri-mir-18a RNA based on footprinting analysis[7]. Regions corresponding to the terminal loop and stem are boxed. The cleavage sites for Microprocessor (Drosha/DGCR8) are indicated by arrowheads. **c** Effect of transiently transfected epitope-tagged T7-hnRNP A1, UP1, UP1-M9, RRM1-M9, and RRM2-M9 in the processing of pri-mir-18a in HeLa cells in culture. Processing of pri-mir-16 was included as a control (white bars). The upper panel shows the level of expression of T7 epitope-tagged hnRNP A1 WT or constructs expressing individual domains (RRM1 or RRM2) or the UP1 fragment (tandem RRM1-RRM2). An M9 sequence was included to direct the nuclear localization of the UP1, RRM1, and RRM2 constructs. **d** Electrophoretic mobility shift assay (EMSA) of UP1 in complex with pri-mir-18a loop and stem RNAs

derived from the pri-mir-18a terminal loop and flanking sequences (Fig. 1b) shows more than 200-fold higher affinity ($K_D$ = 15.5 nM with a 1:1 stoichiometry) to UP1 (Fig. 2a, middle panel). The 12-mer RNA harbors two UAG motifs suggesting that each of these can be recognized by one of the RRMs in a cooperative manner. This is evident from the very large increase in binding affinity compared to the binding of the 7-mer to UP1.

It is remarkable that binding of UP1 to this single-stranded 12-mer RNA is at least 10-fold stronger than binding to full-length pri-mir-18a ($K_D$ = 15.5 nM vs. 147 nM, respectively) (Fig. 2a, middle and right panel, respectively). This may be explained by considering that in the pri-mir-18a stem-loop the second UAG motif is predicted to be base-paired in the loop-proximal stem and thus not freely accessible (Fig. 1b). Binding of UP1 to

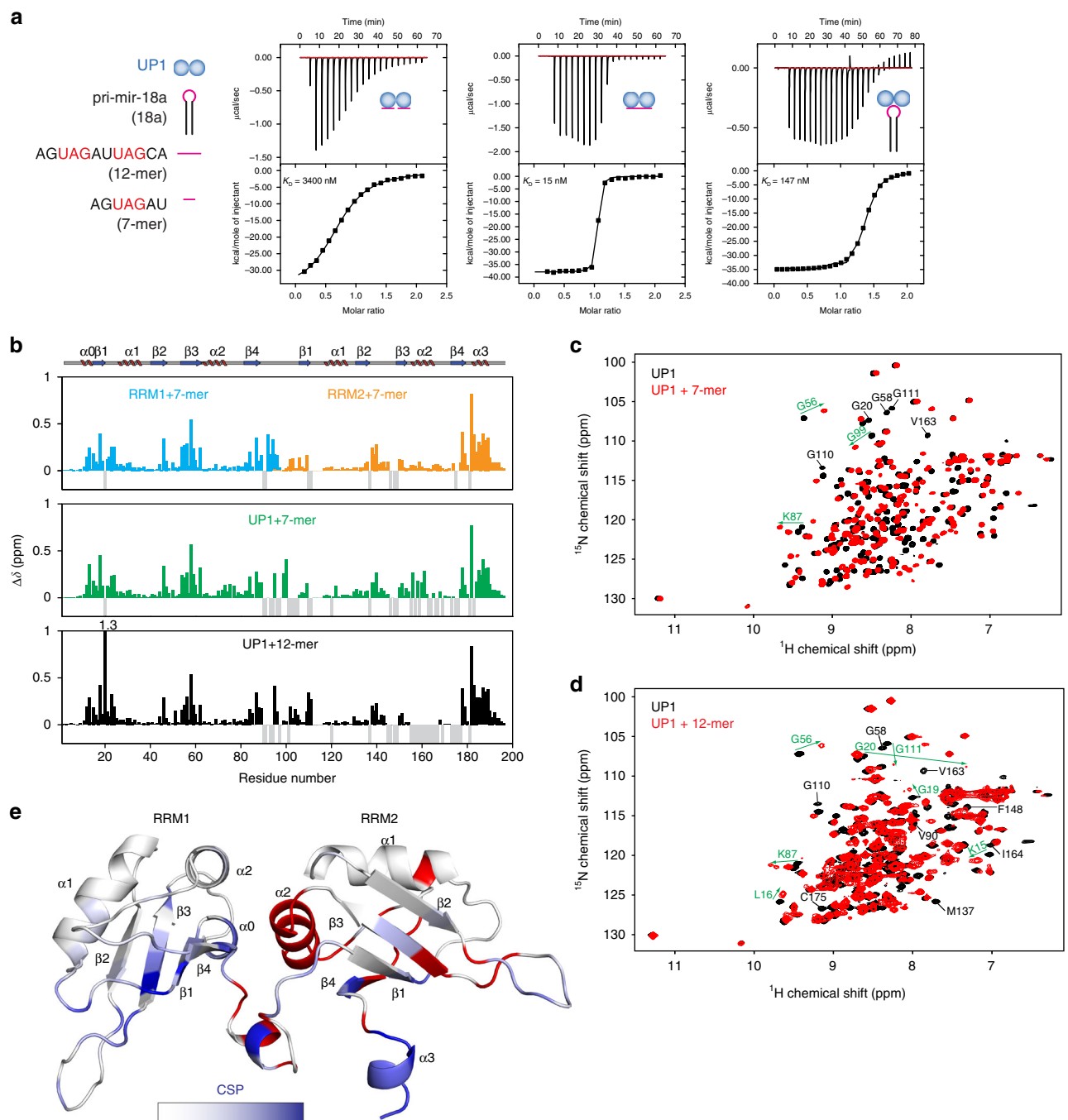

**Fig. 2** Biophysical characterization of UP1–RNA interactions. **a** ITC of the binding of UP1 to 7-mer, 12-mer, and pri-mir-18a RNAs. $K_D$ values are indicated. The sequences of 7-mer and 12-mer oligonucleotides derived from the terminal loop of pri-mir-18a are shown on the left. **b** Combined ${}^1$H and ${}^{15}$N chemical shift perturbations (CSPs) of RRM1/7-mer, RRM2/7-mer, UP1/7-mer, and UP1/12-mer are plotted against the residue number. Secondary structure elements are shown above the plot. The gaps in the graph are proline residues; negative gray bars represent residues that could not be assigned due to line-broadening. **c**, **d** ${}^1$H, ${}^{15}$N correlation spectra of UP1, free (black) and in the presence of either the (**c**) 7-mer or (**d**) the 12-mer RNA (red). Selected residues experiencing large chemical shift perturbations and line-broadening upon RNA binding are labeled in green and black, respectively. **e** The CSPs are mapped onto the structure of UP1 (gradient of white to blue indicates weak to strong CSPs). Residues corresponding to amide signals that are exchange-broadened in the RNA-bound spectra are colored red

**Table 1 Isothermal titration calorimetry of protein–RNA interactions**

| Binding partners | N (protein:RNA) | $K_D$ ($10^{-6}$ M) | $\Delta H$ ($10^4$ cal/mol) | $\Delta S$ (cal/mol/deg) |
|---|---|---|---|---|
| RRM1 + 7-mer (AGUAGAU) | 1.01 | 20.4 ± 1.06 | −1.99 ± 0.07 | −45.2 |
| RRM2 + 7-mer (AGUAGAU) | 1.09 | 6.8 ± 0.14 | −1.64 ± 0.006 | −31.6 |
| UP1 + 7-mer (AGUAGAU) | 0.75 | 3.4 ± 0.12 | −3.53 ± 0.04 | −93.4 |
| UP1 + 17-mer(A35C) | 1.07 | 3.1 ± 0.19 | −1.98 ± 0.03 | −41.2 |
| UP1 + pri-mir-18a | 1.35 | 0.1477 ± 0.0089 | −3.52 ± 0.02 | −86.8 |
| UP1 + 12-mer (AGUAGAUUAGCA) | 1.01 | 0.0155 ± 0.0034 | −3.81 ± 0.02 | −92.0 |
| UP1 + 12-mer-mut1 (AGUUUAUUAGCA) | 1.06 | 0.1541 ± 0.011 | −2.97 ± 0.02 | −68.6 |
| UP1 + 12-mer-mut2 (AGUAGAUUUUCA) | 1.08 | 0.330 ± 0.0265 | −2.45 ± 0.02 | −52.5 |
| UP1 + 12-mer-mut3 (UUUAGAUUAGCA) | 0.98 | 0.00833 ± 0.00135 | −4.10 ± 0.02 | −101 |
| UP1 + 10-mer (UAGAUUAGCA) | 1.07 | 0.01912 ± 0.00673 | −3.94 ± 0.04 | −96.8 |
| UP1(R75E/R88A) + 12-mer (AGUAGAUUAGCA) | 1.08 | 0.0190 ± 0.00205 | −3.84 ± 0.02 | −93.5 |
| UP1(R75E/R88E) + 12-mer (AGUAGAUUAGCA) | 1.1 | 0.0401 ± 0.0044 | −3.68 ± 0.02 | −89.5 |

pri-mir-18a may thus require opening (melting) of these base pairs, whereas in the single-stranded 12-mer RNA both UAG binding sites are readily available for interaction with UP1 (see below), hence the higher affinity. We conclude that the 12-mer RNA is a high-affinity UP1-binding sequence where both RRM domains of UP1 recognize UAG motifs. The RNA recognition features in the 12-mer are expected to represent the interaction of UP1 with the pri-mir-18a. A summary of thermodynamic parameters of the hnRNP A1–RNA interactions is given in Table 1.

**NMR analysis of UP1–RNA interactions**. We next characterized the binding interface of UP1 with various RNA ligands derived from pri-mir-18a using NMR titration experiments. Addition of the 7-mer RNA harboring one UAG motif causes extensive chemical shift perturbations (CSPs) in RRM1 and RRM2 constructs, thus demonstrating that each of the RRMs can interact with the 7-mer RNA (Fig. 2b). The CSP pattern obtained upon titration of the tandem RRM domains in UP1 with the 7-mer RNA is very similar to that obtained for individual RRM domains (Fig. 2b, c). This shows that recognition of the 7-mer RNA in the isolated RRM domains and in the context of the UP1 construct is very similar. Similarly, the 12-mer RNA harboring two UAG motifs induces large CSPs in both RRM domains of UP1 (Fig. 2d) that are comparable to those obtained at saturating levels of the 7-mer. The CSPs map to the canonical RNA-binding surface on the β-sheets of the two RRM domains (Fig. 2e). In addition, strong CSPs are observed for residues in the C-terminal region of RRM2. This region is flexible in free UP1 but upon RNA-binding forms an additional helix (α3), which is not present in the free protein[19] (Supplementary Fig. 3a, b), suggesting that helix α3 is induced and stabilized upon RNA binding. Interestingly, a number of NMR signals that correspond to residues in RRM2 and the RRM1-RRM2 linker are severely broadened in the RNA-bound spectrum, suggesting dynamics on the μs-ms time-scale. The affected residues map to the interface between the RRM1 and RRM2 domains (Fig. 2e), suggesting that some conformational dynamics and adaptation of this interface is associated with RNA binding.

**Recognition of the pri-mir-18a terminal loop by UP1**. To gain insight into the molecular details of pri-mir-18a recognition by UP1, we determined the crystal structure of the UP1/12-mer RNA complex at 2.5 Å resolution (Table 2; Fig. 3a; Supplementary Fig. 4a). Surprisingly, the crystal structure of the UP1/12-mer RNA complex exhibits two molecules of UP1 and two RNA chains in the asymmetric unit in a 2:2 stoichiometry (Supplementary Fig. 4a), similar to a previously reported structure of UP1 with single-stranded telomeric DNA[20]. As this peculiar 2:2 stoichiometry most likely does not represent the UP1:RNA

**Table 2 Crystallographic data collection and refinement statistics**

| | UP1/12-mer |
|---|---|
| *Data collection* | |
| Space group | P6222 |
| Cell dimensions | |
| a, b, c (Å) | 127.03,127.03,147.06 |
| α, β, γ (°) | 90, 90, 120 |
| Resolution (Å) | 48–2.5 (2.6–2.5) |
| $R_{merge}$ | 0.09 (0.98) |
| $I/\sigma I$ | 20.54 (2.09) |
| Completeness (%) | 99.8 (99) |
| Redundancy | 9.9 (9.1) |
| | |
| *Refinement* | |
| Resolution (Å) | 48–2.5 |
| No. reflections | 24,897 |
| $R_{work}/R_{free}$ | 0.20/0.23 |
| No. atoms | 3373 |
| Protein | 2846 |
| RNA | 451 |
| Water | 72 |
| | |
| *B-factors* | |
| Protein | 70.9 |
| RNA | 85.6 |
| Water | 47.01 |
| | |
| R.m.s. deviations | |
| Bond lengths (Å) | 0.008 |
| Bond angles (°) | 1.07 |

The data set is from a single crystal. Values in parentheses are for highest-resolution shell

complex in solution, we analyzed UP1, RNA and the protein–RNA complexes by static light scattering (SLS) (Fig. 3b). Both UP1 and pri-mir-18a alone behave as single species with a molecular weight corresponding to respective monomeric conformations. Importantly, the molecular weight obtained for the UP1/12-mer complex (26.9 kDa) indicates a 1:1 complex (theoretical molecular weight: 26.3 kDa) and demonstrates that the 2:2 stoichiometry observed in the crystal structure is an artifact induced by the crystal environment. Notably, also the molecular weight obtained for the UP1/pri-mir-18a complex (45 kDa) is fully consistent with the formation of a 1:1 complex (Fig. 3a, b).

The crystal structure reveals that each RRM domain in UP1 specifically recognizes one UAG motif in the RNA. Although the stoichiometry does not reflect the solution conformation, the RNA-binding interface is expected to be conserved. This is consistent with the NMR chemical shift perturbations which map to the canonical binding interface, although some minor

differences may exist regarding specific contacts considering the overall different topology. The crystal structure shows that the central AG dinucleotide in each UAG motif is recognized by contacts mainly through conserved RNP motif residues on the β-sheets (Fig. 3c, d), which resembles the recognition of TAG in the UP1-telomeric DNA complex[20] and recently reported structures with RNA[21,22]. Two conserved aromatic residues in RRM1, Phe17 (RNP-2 motif residue located on β1) and Phe59 (RNP-1 motif residue located on β3), are involved in stacking interactions with the bases of A4 and G5, respectively (Fig. 3d). Similarly, in RRM2 Phe108 (RNP-2 motif residue located on β1) and Phe150 (RNP-1 motif residue located on β3) stack with the bases of A9 and G10, respectively. In addition to the stacking interactions

with RNP residues (vide infra), the adenosine is specifically recognized by hydrogen bonds of its exocyclic NH2 group with the main chain carbonyl oxygen of residues Arg88 and Lys179 in RRM1 and RRM2, respectively. A positively charged residue in each domain, Arg55 in RRM1 and Arg146 in RRM2, makes electrostatic interactions with the phosphate backbone of the AG dinucleotide. Two charged residues in each domain, Glu85 and Lys87 in RRM1 and Glu176 and Arg178 in RRM2, contact the uridines in the UAG motifs (U3 and U8), while another charged residue, Lys15 in RRM1 and Lys106 in RRM2, interacts with G5 and G10, respectively (Fig. 3d), thereby recognizing the U and G residues in the UAG motif. Overall, the mode of RNA recognition by RRM1 and RRM2 is very similar; in each domain an AG

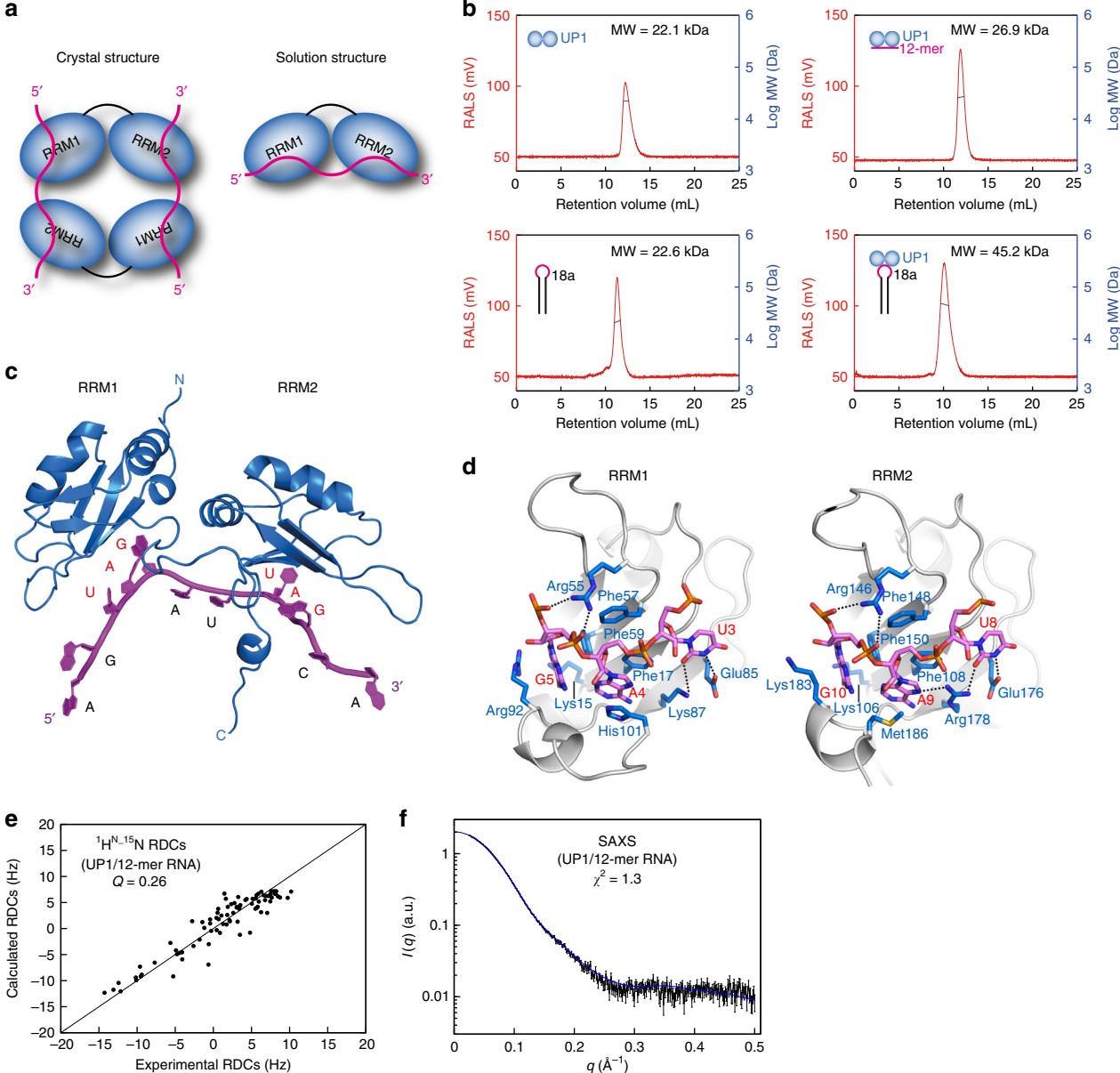

**Fig. 3** Structure of the UP1/12-mer complex. **a** Schematic representation of the 2:2 UP1/12-mer RNA complex observed in the crystal structure (Supplementary Fig. 4a) and the proposed 1:1 complex in solution. **b** Static light scattering (SLS) profiles of UP1, pri-mir-18a, UP1/12-mer, and UP1/pri-mir-18a. The molecular weight (MW) obtained from SLS is indicated. UP1 forms a 1:1 complex with both pri-mir-18a and the 12-mer oligonucleotide. **c** Structural model of the 1:1 UP1/12-mer complex, where each RRM domain recognizes a UAG motif in the 12-mer RNA (magenta) forming a 1:1 complex. **d** Structural details of recognition of the UAG motif by RRM1 and RRM2 domains. Side-chain of selected residues involved in the interaction are indicated. **e, f** Correlation of experimental $^1$H-$^{15}$N residual dipolar couplings (RDCs) (**e**) and small angle X-ray scattering (SAXS) data (**f**) vs. those calculated from the structural model shown in **c**

dinucleotide is sandwiched between the β-sheet surface and a C-terminal helix (Fig. 3d).

The two RRM domains of hnRNP A1 are connected by an approximately 17-residue linker, which is evolutionary conserved both in terms of sequence and length[23] (Supplementary Fig. 3e). Residues in this linker are highly flexible as determined by NMR relaxation data (Supplementary Fig. 3a). However, the two RRM domains interact with each other and this intramolecular domain interface is stabilized by two conserved salt bridges (Arg75–Asp155 and Arg88–Asp157) as well as a small cluster of hydrophobic residues in the interface. Virtually the same domain arrangement and salt bridges are observed in the monomeric solution structure of UP1 free[19] and a crystal structure bound to a short RNA[22] (Supplementary Fig. 4f), indicating that this arrangement of RRM1 and RRM2 stabilized by a number of interactions is preformed and may be retained in free and ligand-bound structures of UP1.

To confirm the domain arrangement of RNA-bound UP1 in solution, we measured NMR and small angle X-ray scattering (SAXS) (Supplementary Fig. 4b–d). NMR paramagnetic relaxation enhancements (PRE) were recorded for UP1 spin-labeled at position 66 (UP1 Glu66Cys mutant). The PRE data provide long-range (up to ~20 Å) distance information and can thus report on domain/domain arrangements[24–26] (Supplementary Fig. 4c). The PRE profiles of free and RNA-bound form of UP1 are similar, suggesting that the domain arrangement does not change significantly in the presence of the 12-mer RNA.

To determine a structural model of the 1:1 UP1/12-mer RNA complex in solution a protein monomer from our crystal structure was used as template. The RNA was initially placed by constraining the recognition of the two UAG motifs in the 12-mer RNA (taken from two different RNA molecules in the 2:2 crystal structure) (Fig. 3a). SAXS and NMR residual dipolar coupling (RDC) data were then used as restraints in a molecular dynamics simulation together with the distances of the Arg75–Asp155 and Arg88–Asp157 salt bridges as observed in the crystal structure. The refinement shows that a 1:1 complex is fully compatible with the data (Fig. 3e, f; Supplementary Table 2). The arrangement of RRM1 and RRM2 is similar to the corresponding interface in the crystal structure (coordinate rmsd of 2.8 Å calculated over all the backbone atoms excluding linker residues) with only minor conformational changes in the domain/domain interface (Fig. 3c; Supplementary Fig. 4a–e).

**Validation of the UP1/12-mer RNA structural model.** The structural model of the 1:1 UP1/12-mer RNA complex was confirmed by mutational analysis of protein and RNA. ITC data with 12-mer RNAs where the first or second UAG motif has been replaced by UUU, 12-mer-mut1: AG*UUU*AU*UAG*CA and 12-mer-mut2: AG*UAG*AU*UUU*CA show 10-fold and 20-fold ($K_D$ = 154 nM and 330 nM) reduced binding affinity, respectively, compared to the wild-type sequence (Table 1; Supplementary Fig. 2b). This confirms that both motifs are recognized by the protein. Additional RNA variants with an AG→UU mutation or lacking the initial AG dinucleotide, that precedes the two "canonical" UAG motifs, have the same binding affinity as the wild-type 12-mer RNA (Table 1; Supplementary Fig. 2b). As mutation or deletion of the AG dinucleotide at the very 5′ end of the 12-mer RNA has essentially no effect, this establishes that UP1 has a preference for the recognition of two closely spaced UAG motifs.

Further, the domain interface in UP1 was probed by mutations that are expected to disrupt the two salt bridges Arg75–Asp155 and Arg88–Asp157 in the RRM1-RRM2 interface, which have been observed in all reported structures of free and nucleic acid

bound forms of UP1[19–22]. Introducing charge clashes (UP1-Arg75Glu/Arg88Glu) in this interface, which is remote from the RNA-binding surface, decreases the binding affinity to the 12-mer RNA by ≈3-fold ($K_D$ ~15.5 nM vs. ~40 nM) (Supplementary Fig. 2c). This suggests that the salt bridges play an indirect role for RNA binding by stabilizing the arrangement of the two RRM domains.

Functionally, we established that mutations of conserved Phe residues in the RNP motifs of RRM2 or both RRMs that affect RNA binding, consequently abrogate the stimulatory effect of UP1-M9 in the processing of pri-mir-18a in HeLa cells in culture (Supplementary Fig. 1a, b; Supplementary Table 1). Collectively, the structural model of the 1:1 UP1/12-mer RNA complex is fully consistent with our biochemical and functional data regarding the requirement for two RRM domains and two UAG motifs for high-affinity interaction.

**UP1 binding destabilizes pri-mir-18a RNA near the terminal loop.** The structure of UP1 bound to the 12-mer RNA in a 1:1 stoichiometry implies that the loop-proximal region of the pri-mir-18a stem should be destabilized to enable recognition of two UAG motifs, one accessible in the terminal loop and the second one as part of the pri-mir-18a duplex. To study this further, we first analyzed the structure of pri-mir-18a alone. For this, a model of the RNA was prepared using the MC-Fold/MC-Sym server[27] (Fig. 4a) and assessed with experimental SAXS and NMR data. The predicted secondary structure and elongated shape of pri-mir-18a 71-mer is supported by SAXS data of the free RNA (Fig. 4b). We then used NMR to analyze the base-pairing in the pri-mir-18a 71-mer using imino NOESY spectra (Fig. 4c), as imino proton NMR signals probe the presence and stability of base pairs[28]. We could unambiguously assign imino-imino cross-peaks corresponding to the stem region of the 71-mer RNA. However, no imino correlation was observed for the upper part of the stem loop (Fig. 4c, see secondary structure on the right), suggesting that this region of the RNA helix is dynamic with less stable base-pairing.

This was further confirmed by analyzing a 17-mer stem-loop construct (Fig. 4d, e), which represents the loop-proximal stem region in the full-length pri-mir-18a (Fig. 1b). The upper stem region in this RNA harbors the second UAG motif that is also present in the single-stranded 12-mer RNA and recognized by UP1. An imino NOESY spectrum of the 17-mer RNA shows imino correlations only for the G:C and G:U base pairs at the stem of this RNA. The fact that the imino protons of the predicted A:U base pairs adjacent to the terminal loop are not detectable indicates that they are dynamic (Fig. 4d). Nevertheless, the presence of these A:U base pairs is confirmed by the (Py) H(CC)NN-COSY experiment, which allows detection of transient and weak-pairs, even in the absence of observable imino proteins[29] (Fig. 4e). The absence of detectable imino protons for many predicted base pairs in the 17-mer and in the upper part of pri-mir-18a indicates that the corresponding stem region exhibits weak base pairs and is dynamic.

The observation of dynamic base-pairing in the loop-proximal region is consistent with melting of the stem region to enable recognition of both UAG motifs by UP1. In fact, NMR titrations of UP1 with the 17-mer stem-loop RNA show virtually identical CSPs to those seen with the single-stranded 12-mer RNA (Fig. 2d, Supplementary Fig. 3c). As NMR chemical shifts are sensitive indicators of the three-dimensional structure, this finding confirms that the protein–RNA interactions observed with the 12-mer RNA also reflect the RNA recognition within the 17-mer RNA. In both cases, two UAG motifs are involved in the

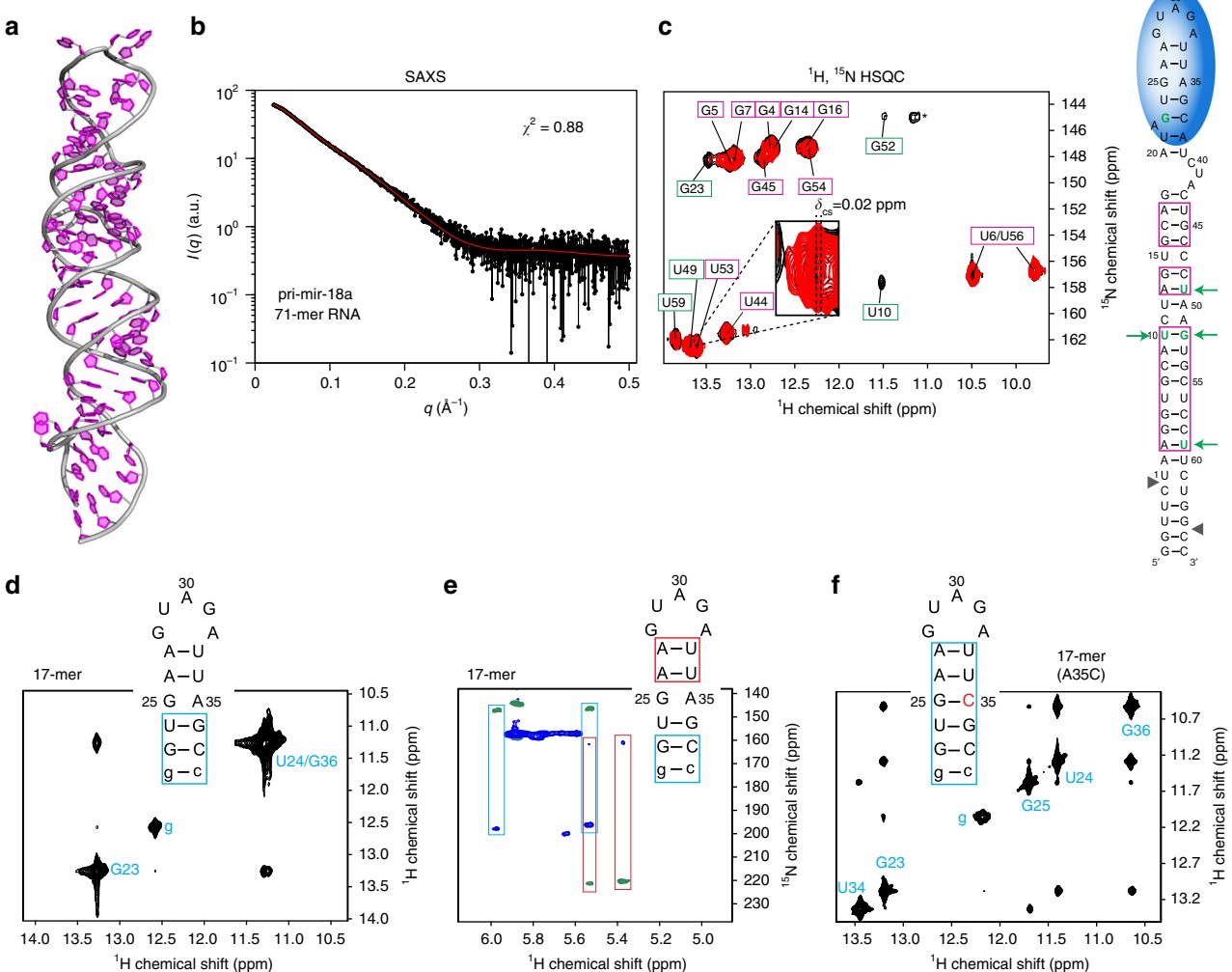

**Fig. 4** Structural analysis of pri-mir-18a 71-mer RNA. **a** A structural model of pri-mir-18a 71-mer RNA obtained using the MC-Fold/MC-Sym server. **b** Experimental and predicted SAXS data for the pri-mir-18a model are shown in black and red, respectively. **c** NMR analysis of the 71-mer pri-mir-18a RNA. $^1$H, $^{15}$N-HSQC of $^{13}$C, $^{15}$N-labeled pri-mir-18a in the absence (black) and presence (red) of UP1. Imino signals observed indicating stable base-pairing are shown on the secondary structure of pri-mir-18a (magenta boxes). The signal marked with an asterisk has not been assigned. Nucleotides undergoing large perturbations and/or line broadening upon UP1 binding are highlighted with green boxes (left panel) or green letters and arrows (right panel). **d** 2D-imino NOESY and **e** H5 correlated HNN experiment of the 17-mer RNA derived from the terminal loop and flanking stem of pri-mir-18a. Base pairs confirmed by these experiments are boxed in blue and red on the secondary structure of the 17-mer RNA. The first G:C base pair, which is not part of the native sequence is shown in lower case. **f** 2D-imino NOESY of the 17-mer(A35C) RNA. The mutated nucleotide is shown in red. Base pairs confirmed by the detection of the imino correlation are boxed

interaction, thus requiring melting of the 17-mer RNA helical stem. This is also consistent with the ITC binding curve (Supplementary Fig. 2d), which shows unusual concentration-dependent enthalpy changes that cannot be fitted to a simple binding event, but may reflect coupled RNA unfolding and protein binding. To further support this, we studied an A35C variant of the 17-mer RNA, which introduces complete Watson–Crick complementarity in the RNA stem. In contrast to the 17-mer, the 17-mer (A35C) exhibits all expected base pairs as evidenced by detectable imino protons throughout the stem region (Fig. 4f). NMR titrations show much smaller chemical shift perturbations in UP1 for the mutant 17-mer RNA (Supplementary Fig. 3d), which binds with a $K_D$ of ~3 μM, i.e., corresponding to a 20–200-fold reduced binding affinity compared to the full-length pri-mir-18a and the single-stranded 12-mer RNA (Table 1; Supplementary Fig. 2d). These findings can be rationalized by considering that in the A35C mutant only one single-stranded UAG motif is available for

binding to UP1, as the second motif is now part of a highly stable RNA helix and thus not accessible. Consistently, the 17-mer (A35C) exhibits micromolar affinity to UP1, i.e., comparable to the affinity observed for the 7-mer RNA with a single UAG motif. On the other hand, the fact that the wild-type pri-mir-18a has one UAG motif in the terminal loop and a second one in the dynamic stem region suggests that binding of UP1 requires melting of the stem region flanking the terminal loop. Taken together, these data suggest that melting of the loop-proximal stem region in pri-mir-18a enables recognition of the second UAG motif and thus provides high-affinity (low nanomolar $K_D$) binding of UP1 to both UAG motifs.

Next, we compared the $^1$H, $^{15}$N imino correlations of the pri-mir-18a 71-mer RNA in the free form and bound to UP1 (Fig. 4c). Notably, signals for the U10:G52 base pair in the middle of the stem region are not detectable in the complex but readily observed in the free form, indicative of destabilization and partial melting of this part of the duplex. Also, other residues especially

next to mismatches and other less stable regions of the RNA exhibit reduced intensity or chemical shift perturbation such as G23, U49, and U59 (Fig. 4c, nucleotides and arrows in green). This is consistent with allosteric effects that lead to destabilization of the complete pri-mir-18a stem-loop induced by binding of UP1 to the terminal loop.

**Structural model of the UP1/pri-mir-18a complex.** To derive a structural model of UP1 bound to pri-mir-18a, we performed molecular dynamics simulations restrained by the structural information obtained for the UP1/12-mer RNA complex and experimental SAXS data of the UP1/pri-mir-18a complex (Fig. 5a; see Methods for details). The resulting structure was then scored

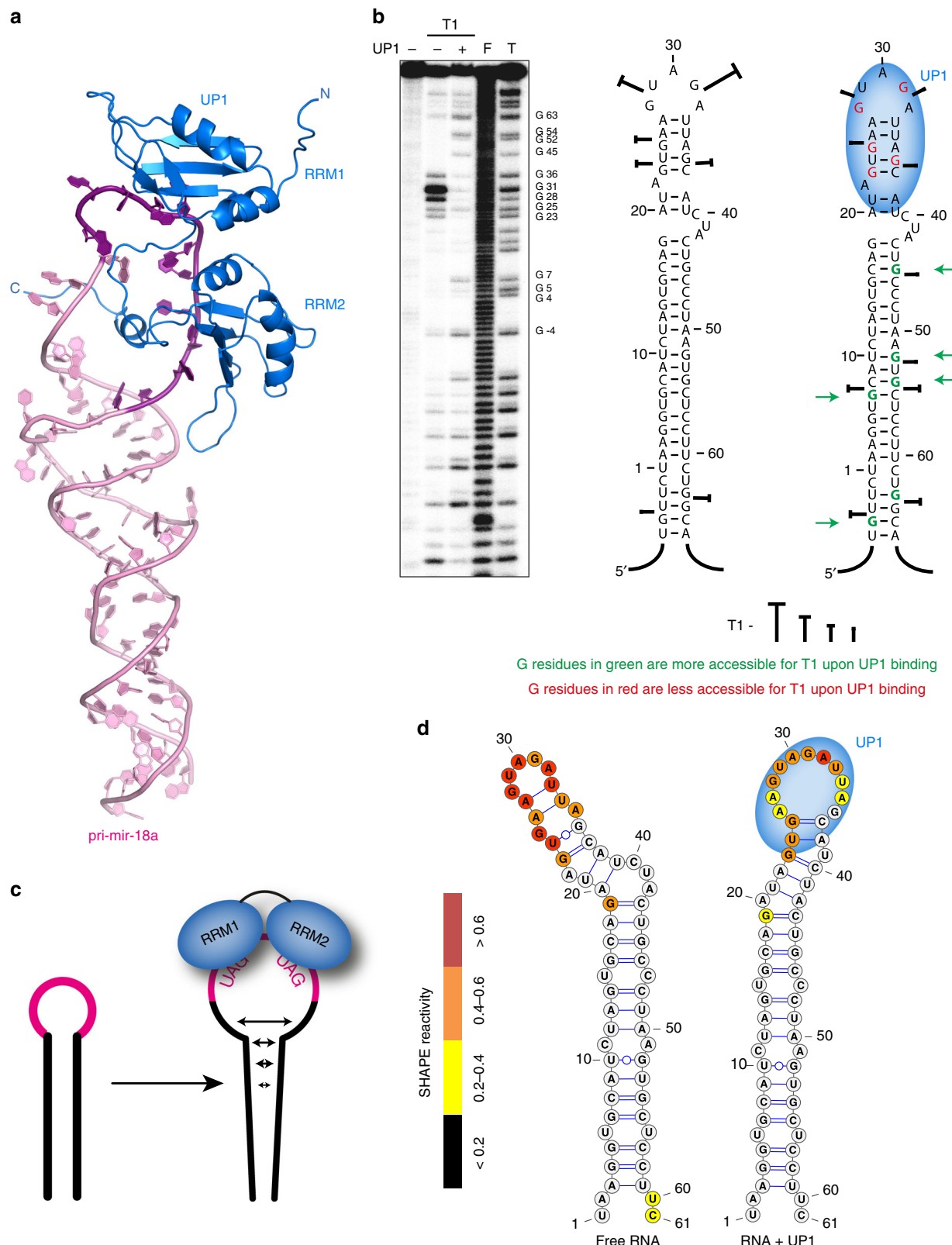

against a combination of small angle neutron scattering (SANS) data with contrast matching (Fig. 5a; Supplementary Fig. 4g, h). SAXS data show that the dimensions of the complex correspond to a radius of gyration ($R_g$) and maximum pairwise distance ($D_{max}$) values of 37.7 Å and 130 Å, respectively, consistent with a 1:1 complex (Supplementary Table 2). The structural model shows very good agreement with all experimental data and is also consistent with ab initio SAXS derived models of the protein–RNA complex (Supplementary Fig. 4g, h). The UP1/pri-mir-18a complex shows the recognition of two UAG motifs in the terminal loop and a partially melted upper stem of pri-mir-18a (Fig. 5a).

To validate the structural model described and to assess changes in accessibility of the pri-mir-18a induced by UP1 binding, we performed foot-printing analysis of the complete pri-mir-18a RNA in the absence and presence of UP1 (Fig. 5b). This revealed that in the free RNA the terminal loop and flanking stem region comprising the two UAG motifs are accessible and dynamic (Fig. 5b), which is consistent with the NMR data (Fig. 4c, d). Binding of UP1 protects this region in the RNA, in full agreement with the structural model of the UP1/pri-mir-18a complex. The significant reduction in accessibility observed for residues in the terminal loop is consistent with the simultaneous binding of both RRM domains to the RNA (Figs. 3c and 5a). Interestingly, residues at the bottom part of the stem of pri-mir-18a become more accessible for nuclease cleavage upon binding of UP1 to the terminal loop (Fig. 5b, green nucleotides). This is fully consistent with the NMR analysis which indicates that UP1 binding leads to destabilization of the pri-mir-18a stem (Fig. 4c).

**SHAPE analysis of pri-mir-18a.** To assess the RNA structure of pri-mir-18a and potential effects of flanking regions in the context of the pri-mir-17-18a-19a cluster, we performed selective 2′-hydroxyl acylation analyzed by primer extension (SHAPE, Supplementary Fig. 5a, b)[30]. In vitro-transcribed RNAs comprising either pri-mir-18a or the pri-mir-17-18a-19a cluster were incubated with increasing amounts of purified UP1 or full-length hnRNP A1 proteins, prior to treatment with N-methylisatoic anhydride (NMIA). The SHAPE reactivity reflects the average intensity of the NMIA-treated RNA primer extension products, normalized to the corresponding untreated RNA, in the presence or absence of UP1/hnRNP A1 proteins.

SHAPE analysis of pri-mir-18a reveals a region of medium to high nucleotide flexibility, corresponding to the terminal loop (nts 170–186). As indicated by NMR (compare Fig. 4d, e), this region of the pri-mir-18a stem loop is dynamic, consistent with varying degrees of SHAPE reactivity for the loop-proximal base pairs, thus supporting the dynamic nature of the upper stem loop. The secondary structure of the free pri-mir-18a was predicted using both SHAPE reactivity and NMR-identified base pairs with RNAstructure[31] (Fig. 5d). The structure is in excellent agreement with the SAXS, enzymatic probing, and NMR data.

SHAPE analysis of pri-mir-17-18a-19a revealed few differences between SHAPE reactivity of pri-mir-18a in the cluster and isolated pri-mir-18a, indicating that the pri-mir-18a hairpin folds largely independently of the rest of the cluster (Supplementary Fig. 5b). Despite the overall similar reactivity pattern, differences observed from nucleotide 210 through 220 are presumably induced by the presence of pri-mir-17 and pri-mir-19a in the whole transcript, which may stabilize the basal AU base pairs of pri-mir-18a. These results are corroborated by a recent SHAPE and chemical probing assays performed on the complete oncomiR-1 cluster[32].

Importantly, upon addition of UP1 and hnRNP A1, distinct changes in SHAPE reactivities are observed compared to the free pri-mir-18a indicating that residues around the terminal loop and flanking stem region (nts 170–186) become protected by protein binding (Fig. 5d; Supplementary Fig. 5a). This region includes the two UAG motifs that bind to RRM1 and RRM2 of hnRNP A1/UP1. Analysis of individual replicate SHAPE data sets reveals a modest increase in SHAPE reactivity down the RNA stem in the presence of protein (nts 50, 51, 53–55). However, these changes are not visible after averaging the three SHAPE replicate data sets, and are thus not statistically significant. While SHAPE may not be sensitive enough to detect the destabilization of the central stem upon hnRNP A1 binding, the SHAPE reactivity observed supports the proposed secondary structure and a plethora of evidence indicating that hnRNP A1 binds to and opens the terminal loop.

Changes in SHAPE reactivity observed for the pri-mir-17-18a-19a cluster in the presence of hnRNP A1/UP1 also revealed the protection of nucleotides spanning the terminal loop of pri-mir-18a (nts 170–186). Thus, most of the highly reactive residues in pri-mir-18a display a similar behavior as the pri-mir-17-18a-19a transcript upon addition of UP1/hnRNP A1 proteins (Fig. 5d; Supplementary Fig. 5). Importantly, both UP1 and hnRNP A1 induce comparable changes in SHAPE reactivity, confirming that UP1 is sufficient to bind to the pri-miRNA, in agreement with the EMSA and functional assays (Fig. 1c, d).

In summary, relative to the free RNA the presence of hnRNP A1/UP1 leads to a decreased SHAPE reactivity around the terminal loop of pri-mir-18a, indicating that this region is the major binding site for the RRMs of hnRNP A1.

**Mechanism of hnRNP A1 stimulation of pri-mir-18a processing.** Finally, we attempted to address the mechanism by which hnRNP A1 activates the processing of pri-mir-18a. One possible scenario is that binding of hnRNP A1 (or UP1) leads to partial opening/melting of the terminal loop, which can lead to destabilization of the stem region and thus render it more accessible for processing by Drosha[7]. Indeed, footprinting and site-directed mutagenesis of pri-mir-18a suggested that hnRNP A1 alters the local conformation of the stem in the vicinity of Drosha cleavage sites[7], although the molecular mechanism was unclear. Indeed,

**Fig. 5** hnRNP A1 recognizes the terminal loop of pri-mir-18a. **a** Structural model of the 71-mer pri-mir-18a/UP1 complex. The region corresponding to the 12-mer RNA in the terminal loop of the pri-mir-18a is shown in dark magenta. **b** Footprint analysis of the pri-mir-18a/UP1 complex. Cleavage patterns were obtained for 5′ $^{32}$P-labeled pri-mir-18a transcript (100 × 10$^3$ c.p.m.) incubated in the presence of recombinant UP1 protein (+200 ng, 500 nM), treated with Ribonuclease T1 at 1.5 U/µL. F and T identify nucleotide residues subjected to partial digest with formamide (every nucleotide) or ribonuclease T1 (G-specific cleavage), respectively. The cleavages intensities generated by Ribonuclease T1 are indicated on the pri-mir-18a secondary structure. The region of the major UP1 footprints is indicated by a blue oval shape. Residues that show enhanced reactivity upon UP1 binding are indicated by green arrows. **c** A schematic model of the effects of hnRNP A1 on pri-mir-18a. Binding of UP1 to the terminal loop, where each RRM domain recognizes a UAG motif, leads to melting of the terminal loop that then spreads down leading to destabilization and enhanced dynamics in the RNA stem, thereby facilitating Drosha cleavage by unknown mechanisms. **d** Averaged SHAPE and NMR data were used to calculate the secondary structure of pri-mir-18a free and when bound to UP1. SHAPE data were implemented in RNAstructure[31] as a soft constraint, while base pairs identified by NMR were used as hard constraints. The structures shown correspond to the lowest free energy models. Color coding corresponds to SHAPE reactivity as indicated with red/orange/yellow corresponding to high/semi/low reactivity

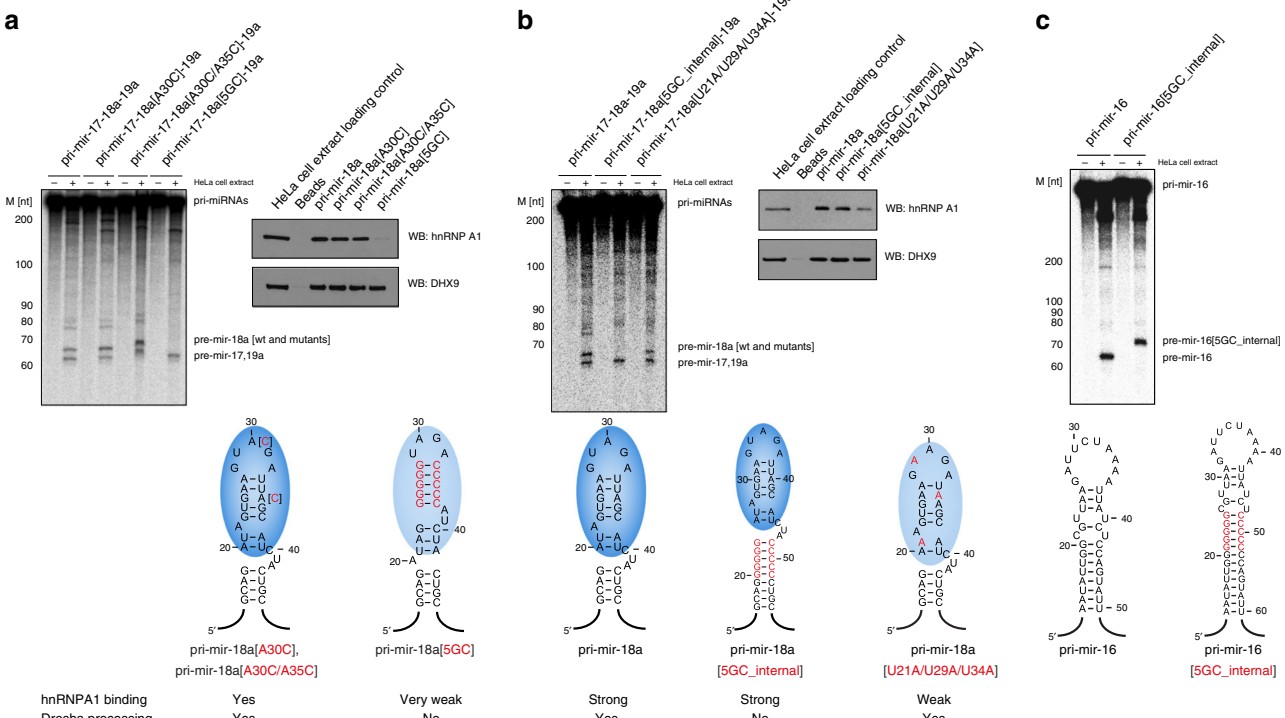

**Fig. 6** Mechanism of hnRNP A1 stimulation of pri-miRNA processing. **a** Pri-mir-18a with single or double A→C point mutations still bind hnRNP A1 in RNA pull-down assays and are processed by Drosha (in vitro processing assay with the pri-mir-17-18a-19a cluster, wildtype and mutants), whereas pri-mir-18a mutant with a 5GC clamp does not bind hnRNP A1 (RNA pull-down assay is shown on the right) and is not processed by Drosha. **b** Pri-mir-18a with a 5GC_internal clamp and wild-type terminal loop binds hnRNP A1 but is not processed by Drosha. Pri-mir-18a with triple mutations [U21A/U29A/U34A] binds hnRNP A1 with lower affinity than the wild-type pri-mir-18a but is still efficiently processed by Drosha. **c** In vitro processing assay of pri-mir-16 with 5GC_internal clamp shows efficient processing by Drosha, similar to pri-mir-16 wildtype

UP1 (unwinding protein 1) can unwind secondary and higher order structures of DNA and RNA[15,33]. We constructed a series of mutants in the terminal loop region of pri-mir-18a. These include single and double nucleotide mutants, where an A residue within the UAG motif in the terminal loop or within the second UAG motif, were mutated to C (UCG in pri-mir-18a [A30C] and pri-mir-18a[A30C/A35C], respectively), and a triple mutant (pri-mir-18a[U21A/U29A/U34A]), where all UAG motifs were mutated to AAG. We also designed a mutant, in which the terminal loop was stabilized by five G:C base pairs (pri-mir-18a[5GC]). The wild-type sequence was efficiently processed in the context of the pri-mir-17-18a-19 cluster (Fig. 6a) and isolated pri-mir-18a (Supplementary Fig. 6). The single (pri-mir-18a [A30C]) and double nucleotide (pri-mir-18a[A30C/A35C]) terminal loop mutant RNAs retained hnRNP A1 binding, although with lower affinity. These mutant RNAs were still efficiently processed in the context of the pri-mir-17-18a-19 cluster (Fig. 6a) and the isolated pri-mir-18a (Supplementary Fig. 6). The triple mutant (pri-mir-18a[U21A/U29A/U34A]) showed binding to hnRNP A1, although with reduced affinity, and retained efficient processing in the pri-mir-17-18a-19 cluster (Fig. 6b). However, it displayed decreased processing in the isolated pri-mir-18a (Supplementary Fig. 6). The pri-mir-18a[5GC] mutant does not bind to hnRNP A1 in the RNA pull-down assay and consequently is not processed (Fig. 6a; Supplementary Fig. 6). This lack of processing could result from disruption of UP1 binding and/or conformational changes that inhibit Microprocessor activity, for example, by stabilizing the loop-proximal stem region. Based on these data we conclude that hnRNP A1 binding is essential for pri-mir-18a processing, although some reduction in binding affinity can still support miRNA processing,

as shown by the fact that even low hnRNP A1 levels are sufficient to stimulate processing activity.

Next, we hypothesized that destabilization and unfolding of the upper stem region close to the terminal loop in pri-mir-18a induced by hnRNP A1 binding can spread along the RNA stem and thus lead to destabilization of the RNA near the Drosha cleavage site. To examine this hypothesis, we tested a different mutant pri-mir-18a, in which the terminal loop was clamped by 5 G:C base pairs (pri-mir-18a[5GC_internal]). Note, that this RNA variant is different from the pri-mir-18a[5GC] mutant described above, in which the terminal loop region was stabilized by 5 G:C base pairs. Remarkably, the processing of this RNA was impaired in the context of pri-mir-17-18a-19 cluster (Fig. 6b) and isolated pri-mir-18a (Supplementary Fig. 6), despite retaining full binding to hnRNP A1 in the RNA pull-down assay (Fig. 6b). This is highly important, because it demonstrates that abrogating the allosteric change induced by hnRNP A1 binding leads to lack of stimulatory processing activity, even in the presence of high-affinity binding.

To rule out the possibility that the internal 5GC clamp affects Drosha processing irrespective of the requirement for hnRNP A1 as an auxiliary factor, we examined the processing of pri-mir-16 with and without the 5GC clamp. Both wild-type pri-mir-16 and pri-mir-16[5GC_internal] were processed by Drosha, indicating that the internal 5GC clamp is not sufficient to impair Drosha processing (Fig. 6c). Furthermore, to control for undesired effects of our mutations on Drosha cleavage we generated a set of pri-mir-16 chimerae that harbor wild-type or mutated pri-mir-18a terminal loops (Fig. 7). Drosha efficiently processes all but one of the pri-mir-16/18a chimerae analyzed here, where only the processing of pri-mir-16/18a[5GC] was reduced. This shows that

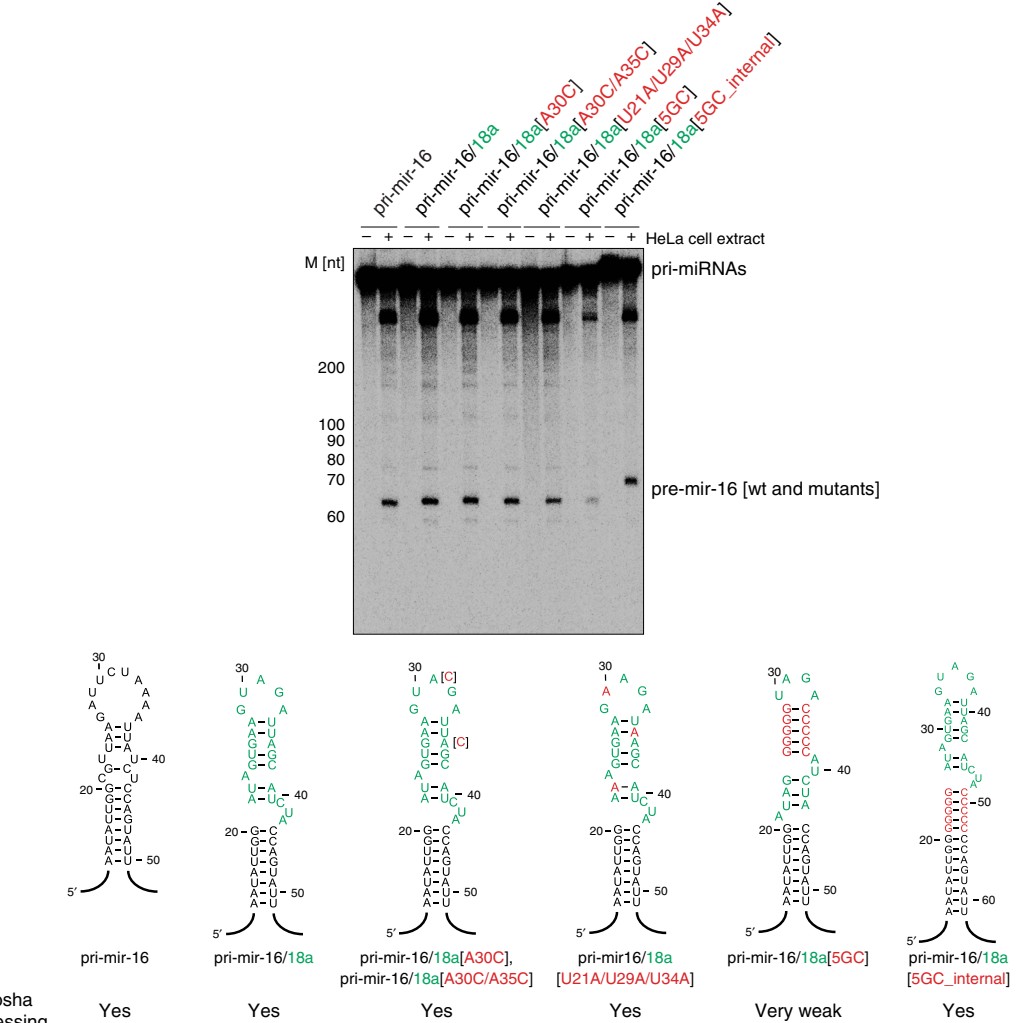

**Fig. 7** In vitro processing of pri-mir-16 and 18a chimerae. Wild-type pri-mir-16 or chimerae with 18a terminal loop sequences (in green) with point mutations introduced in Supplementary Fig. 6 (in red) were subjected to in vitro processing assay. All transcripts, except for pri-mir-16/18a[5GC], are efficiently processed by Drosha. Pri-mir-16/18a[5GC] processing efficiency is reduced, indicating that conformational changes to the pri-mir stem can directly affect Microprocessor activity

conformational changes in the apical part of the pri-mir stem can directly affect Microprocessor activity. By contrast, the processing of pri-mir-16/18a[5GC_internal]) was efficient in the context of this chimera (Fig. 7), unlike what we observed with the processing of pri-mir-18a[5GC_internal], which was abolished (Fig. 6b). Therefore, these data strongly support our hypothesis that binding of hnRNP A1 to the terminal loop of pri-mir-18a results in a partial unwinding of this pri-miRNA that spreads from the terminal loop towards the stem and is essential for the stimulatory role of hnRNP A1 in miR-18a biogenesis.

## Discussion
Here, we have used a multi-disciplinary approach to reveal the molecular mechanism by which hnRNP A1 binds to pri-mir-18a and facilitates its processing. We show that hnRNP A1 specifically binds to pri-mir-18a through interactions involving both RRM domains in UP1 and a region comprising two UAG RNA sequence motifs in the terminal loop of pri-mir-18a and a flanking stem region. Cooperative binding of both domains to cognate RNA motifs results in substantially increased binding affinity and allows the unwinding of the target stem-loop

RNA. This mode of binding is distinct from the recognition of a viral RNA, where apparently only RRM1 is involved in the recognition of a single AGU motif[22], but reminiscent of recognition of two RNA motifs in single-stranded cis regulatory elements in alternative splicing regulation by hnRNP A1[21]. A common feature for all structures is that the overall domain arrangement and interface in free and nucleic acid bound hnRNP A1 tandem RRMs is conserved (Supplementary Fig. 4f). Nevertheless, some adaptation and fine-tuning of the domain arrangement occurs upon RNA binding. This is consistent with the line-broadening of amide signals in the RRM1/RRM2 interface upon binding to the 12-mer RNA (Fig. 2e). A further common feature is the formation of 1:1 complexes in solution, very distinct from the 2:2 stoichiometry observed in crystal structures. The various distinct modes of nucleic acid recognition by hnRNP A1 are intriguing and may reflect how it can play important roles in the regulation of many distinct biological activities by its RNA interactions.

hnRNP A1 can also act as a negative regulator of let-7a processing, by competing with the activator protein KSRP for binding to the pri-let-7a terminal loop[34,35]. In addition to hnRNP A1, several other RBPs recognize the terminal loop of miRNA precursors and influence, either positively or negatively, their

biogenesis at the post-transcriptional level[36]. For example, binding of Lin28 to the terminal loop of pre-let-7 was shown to inhibit processing by both Drosha and Dicer[37]. Another example is the RBP Rbfox, which binds to the terminal loop of a subset of pri-miRNAs and suppresses their nuclear processing[38].

How does binding of a regulator to the terminal loop of pri-miRNA affect Drosha cleavage at the opposite end of the RNA? We propose that binding of hnRNP A1 to the terminal loop of pri-mir-18a leads to destabilization of base pairs and (partial) melting of the loop-proximal stem region, with subsequent spreading of destabilization of the stem region towards the Drosha cleavage site (Fig. 5c). In support of this model, processing of a mutant pri-mir-18a, in which the terminal loop was clamped by 5 G:C base pairs (5GC_internal) was abolished, despite strong hnRNP A1 binding (Fig. 6b). This strongly argues that the effect of hnRNP A1 binding at the terminal loop is somehow propagated and leads to a stimulatory effect of Drosha processing. Indeed, pri-mir-18b, which is part of the homologous miR-106a-363 cluster, does not require hnRNP A1 for efficient processing[8]. Mechanistically, this can be explained by the fact that the conformation of the stem in pri-mir-18b resembles the more open stem structure comprising a bulge in the stem, which is only observed in pri-mir-18a upon binding of hnRNP A1[7,8]. This affects Drosha processing as simply introducing this bulge in the pri-miR-18a stem (UC→GU) made its processing more efficient and completely independent of hnRNP A1[7,8].

We propose three non-mutually exclusive mechanisms by which hnRNP A1 may enhance the processing of pri-mir-18a. hnRNP A1 binding may 1) promote Drosha processing by stem destabilization and thereby facilitate the recruitment and/or assembly of an active Microprocessor substrate complex, 2) enhance the assembly of pri-mir-18a with the correct orientation onto the Microprocessor by generating a thermodynamic or conformational asymmetry in the stem, 3) increase the accessibility of the Microprocessor to the pri-mir-18a hairpin in the context of the complete oncomiR-1 cluster. Options 1 and 2 are consistent with recent biochemical, structural and computational studies that have shown that the Microprocessor recognizes two regions at either end of pri-miRNA, i.e., near the ssRNA/stem basal junctions at the bottom and near the terminal loop junction[39–44]. Recognition of specific sequences (a UG motif 5′ or CNNC motif in single-stranded region 3′ of the basal junction, and/or a UGUG motif near the terminal loop) and/or structural features at these two opposite regions of a pri-miRNA are proposed to guide the binding orientation and conformation of the Drosha/DGCR8 complex and define the cleavage site. Thus, various RBPs that recognize specific sequence motifs in terminal loops may further regulate and enhance processing of specific pri-miRNAs.

Here, we found that strengthening the upper part of pri-mir-18a stem by GC base pairs blocks miRNA processing, whereas disruption of the base pairs enhances Microprocessor cleavage efficiency. It is noteworthy, that the destabilization of the apical RNA helix by binding of hnRNP A1 induces a structural and thermodynamic asymmetry in the stem region (stably base-paired vs. dynamic RNA helix) that may help define, in which orientation pri-mir-18a is recognized by the Microprocessor. It was recently proposed that heme binding by DGCR8 is required to ensure the correct Microprocessor assembly[41]. A similar role may thus extend to RBPs such as hnRNP A1 that bind to terminal loops of specific pri-miRNAs.

It is also likely that the processing efficiency of pri-mir-18a is context-dependent, suggesting that the sequence and/or structure of pri-mir-18a as part of the miR-17–92 cluster is not optimal for

Drosha processing[8]. Interestingly, several studies have shown that the miR-17–92 cluster adopts a compact tertiary structure, in which individual miRNAs have different expression levels depending on their accessibility[32,45,46]. Notably, a recent SHAPE analysis of the miR-17–92 cluster revealed that the terminal loop of pri-mir-18a in the cluster is partially solvent inaccessible[32]. As the terminal loop corresponds to the sequence that we have identified as the main hnRNP A1-binding site in pri-mir-18a, it is tempting to speculate that binding of hnRNP A1 to the miR-17–92 cluster is associated with a conformational change in the RNA, which can facilitate Microprocessor assembly and Drosha cleavage. We propose that the tertiary structure of pri-mir-18a in the context of the miR-17–92 cluster, as well as sequences in the stem and loop region, are important determinants of miRNA processing by Drosha. The recognition of the pri-mir-18a stem-loop by hnRNP A1 thus adds an additional layer for the regulation of pri-miRNA processing by an RBP, beyond features that have been recently identified[39,43,47].

In conclusion, our data demonstrate that recognition of a conserved terminal loop RNA sequence in pri-miRNAs by an RBP can strongly modulate miRNA biogenesis by conformational changes and dynamic destabilization induced by RNA binding. This suggests that recognition of pri-miRNAs by RBPs is a general paradigm for context-dependent regulation of miRNA biogenesis and function.

## Methods

**Protein expression and purification**. The sequences encoding RRM1 (amino acids 1–97), RRM2 (amino acids 94–196), and UP1 (amino acids 1–196) were subcloned into the pETM11 vector (EMBL) encoding a His tag followed by TEV cleavage site (primers are listed in Supplementary Table 3). Recombinant proteins were expressed in *E. coli* BL21(DE3) cells (Novagen) in standard media or minimal M9 media supplemented with 1 g/L $^{15}$N-ammonium chloride and 2 g/L $^{13}$C-glucose. Protein expression and purification was done as described previously[10]. After growth of bacterial cells up to an OD$_{600}$ of 0.7–0.8, protein expression was induced by 0.5 mM IPTG and continued overnight at 20 °C. Cells were resuspended in buffer A (30 mM Tris/HCl pH 7.5, 500 mM NaCl, 10 mM imidazole, 1 mM TCEP, 5% (v/v) glycerol) supplemented with protease inhibitors, and lysed by sonication. The cleared lysate was loaded on Ni-NTA resin and after several washing steps with buffer A and buffer A containing 25 mM imidazole the protein was eluted with elution buffer containing 300 mM imidazole. After cleavage of the tag by His-tagged TEV protease at 4 °C overnight, samples were reloaded on Ni-NTA resin to remove the tag, TEV protease and uncleaved protein. All protein samples were further purified by size-exclusion chromatography on a HiLoad 16/60 Superdex 75 column (GE Healthcare) equilibrated with NMR buffer (20 mM sodium phosphate pH 6.5, 100 mM NaCl, 2 mM DTT).

**Expression vectors**. The plasmid pCGT7 hnRNP A1 and pCG T7 UP1 have been previously described[48]. In brief, these expression vectors are under the control of the CMV enhancer/promoter and include an N-terminal T7 epitope tag that corresponds to the first eleven residues of the bacteriophage T7 gene 10 capsid protein. The construct pCG T7 UP1-M9 harbors the M9 import/export sequence fused downstream of both RRM domains of UP1. The UP1-M9 mutants were synthesized by Invitrogen including the corresponding mutations and subcloned into pCG T7 (Supplementary Table 1). The mutated residues correspond to F148D/F150D (UP1-M9_FD2); F57D/F59D/F148D/F150D (UP1-M9_FD12); F17D/F57D/F59D/F108D/F148D/F150D (UP1-M9_FD12a); F17A/F57A/F59A/F108A/F148A/F150A (UP1-M9_FA12b).

**Indirect immunofluorescence**. Cells were fixed 24 h after transfection with 4% p-formaldehyde in PBS for 15–30 min at room temperature and permeabilized for 10 min in 0.2 % Triton X-100. Cells were later incubated with 1:1000 anti-T7 monoclonal antibody (Merck, 69522) for 1 h at room temperature, washed with PBS, and further incubated with 1:200 fluorescein-conjugated goat anti-mouse IgG (Cappel Laboratories) (1 h at room temperature). Samples were observed on a Zeiss Axioskop microscope and the images were acquired with a Photometrics CH250 cooled CCD camera using Digital Scientific Smartcapture extensions (IP Lab Spectrum software). The immunofluorescence figures show representative data, and each experiment was reproduced in multiple independent transfections.

**Analysis of miRNA levels in living cells**. HeLa cells were cultured in standard DMEM medium (Life Technologies) supplemented with 10% FBS (Life Technologies). Plasmids were transfected into HeLa cells using Lipofectamine 2000 reagent, as described before[49]. Mouse monoclonal anti-T7 tag HRP conjugate (1:10000, 69048, RRID—AB1080747495, Novagen) was used to detect T7-tagged proteins. Mouse monoclonal anti-tubulin (1:10000, T6199, RRID—AB_477583, Sigma-Aldrich) was used as loading control. miScript qRT-PCR kit (Qiagen) was used to detect miRNAs in total RNA isolated with TRIzol reagent (Life Technologies). Each sample was run in duplicate. To measure the levels of the corresponding miRNAs, values were normalized to 5 S RNA. For each measurement, three independent experiments were performed.

**RNA samples**. Unlabeled 7-mer, 10-mer, 12-mer, and 17-mer RNA oligonucleotides were purchased from IBA in double-desalted form (Göttingen, Germany). All other RNA samples were made by in vitro transcription and purified by denaturing PAGE. RNA samples were heated to 95 °C for 3 min and snap-cooled on ice before use to promote proper folding.

**EMSA**. Electrophoretic mobility shift assays (EMSA) were performed with internally labeled RNAs and proteins produced in *E. coli*. Gel-purified RNAs (50 × $10^3$ c.p.m. (counts per minute), ~20 pmol) were incubated in 15 μL reaction mixtures containing the specified amounts of proteins in Roeder D buffer (100 mM KCl, 20% (v/v) glycerol, 0.2 mM EDTA, 100 mM Tris at pH = 8.0, 0.5 mM DTT, 0.2 mM PMSF) complemented with 0.5 mM ATP, 20 mM creatine phosphate, and 3.2 mM MgCl$_2$. Reactions mixtures were incubated at 4 °C for 1 h followed by electrophoresis on a 6% (w/v) non-denaturing gel. The radioactive signal was captured with radiographic film or was exposed to a phosphoimaging screen and scanned on a FLA-5100 scanner (Fujifilm).

**In vitro processing assays**. Pri-miRNA substrates were obtained by in vitro transcription with [α-$^{32}$P]-UTP. Gel-purified substrates (20 × $10^3$ c.p.m. (counts per minute), ~20 pmol) were incubated in 30 μL reaction mixtures containing 50% HeLa cell extract in Roeder D buffer, 0.5 mM ATP, 20 mM creatine phosphate, and 3.2 mM MgCl$_2$. Then the reactions were incubated at 37 °C for 30 min. The reactions were stopped with 2 × Urea Dye and followed by 8% (w/v) denaturing gel electrophoresis. The signal was registered with a radiographic film or by exposure to a phosphoimaging screen and scanning on a FLA-5100 scanner (Fujifilm).

**RNA pull-down**. RNA pull-down was performed as previously described[49]. In brief, protein extracts from HeLa cells were incubated with in vitro-transcribed RNAs, coupled to adipic acid dihydrazide-agarose beads (Sigma). Next, the beads were washed three times with buffer G (20 mM Tris pH 7.5, 135 mM NaCl, 1.5 mM MgCl$_2$, 10% (v/v) glycerol, 1 mM EDTA, 1 mM DTT, and 0.2 mM PMSF). After the final wash, the proteins associated with the beads were resolved and identified by SDS–PAGE and western blotting, respectively. The following antibodies were used: rabbit polyclonal anti-DHX9 (1:1000, 17721-1-AP, RRID—AB_2092506, Protein-Tech); rabbit polyclonal anti-hnRNP A1 antibody (1:1000, PA5-19431, Invitrogen).

**Footprinting assays**. The assays were performed as described before[50]. In summary, the RNA substrates were synthesized by in vitro transcription and were 5′ labeled with PKA. A formamide ladder and ribonuclease T1 ladder were generated to assign the nucleotide position. Ribonuclease T1 at 0.5 U/μL was added to 1 μL of RNA (50 × $10^3$ c.p.m.) and 7 μL of 1 × structure buffer (12 mM Tris-HCl at pH = 7.5, 48 mM NaCl, 1.2 mM MgCl$_2$). RNAs substrates were unfolded at 90 °C for 1 min and left at RT for 5 min to refold. The footprinting reactions were incubated at 37 °C for 10 min. Reactions were run in the presence and absence of the recombinant UP1 protein and ribonuclease T1. Reactions were run on 10% polyacrylamide gel. The signal was registered with a radiographic film or via exposure to a phosphoimaging screen and then scanned on a FLA-5100 scanner (Fujifilm).

**NMR spectroscopy**. NMR experiments were recorded at 298 K on 900, 800, and 600 MHz Bruker Avance NMR spectrometers, equipped with cryogenic triple resonance gradient probes. NMR spectra were processed by NMRPipe[51] and analyzed using Sparky (T. D. Goddard and D. G. Kneller, SPARKY 3, University of California, San Francisco). Samples were measured in NMR buffer with 10% D$_2$O added as lock signal. Backbone resonance assignments of UP1 alone and in complex with RNA were obtained from a uniformly $^{15}$N,$^{13}$C-labeled UP1 (with random fractional deuteration) in the absence and presence of saturating concentrations of RNA. Standard triple resonance experiments HNCA, HNCACB and CBCA(CO)NH[52] were recorded at 600 MHz.$^{15}$N relaxation experiments were recorded on a 600-MHz spectrometer at 25 °C. $^{15}$N $T_1$ and $T_{1\rho}$ relaxation times were obtained from pseudo-3D HSQC-based experiments recorded in an interleaved fashion with 12 different relaxation delays (21.6, 86.4, 162, 248.4, 345.6, 432, 518.4, 669.6, 885.6, 1144.8, 1404, and 1782 ms) for $T_1$ and 8 different relaxation delays (5, 7, 10, 15, 20, 25, 30, and 40 ms) for $T_{1\rho}$. Two delays in each experiment were recorded in duplicates for error estimation. Relaxation rates were extracted by

fitting the data to an exponential function using the relaxation module in NMRView[53].

Paramagnetic relaxation enhancements (PREs) were recorded at 600 MHz using a sample with concentration of ~200 μM. UP1(Glu66Cys) was spin-labeled by adding 10-fold excess of 3-(2-Iodoacetamido)-PROXYL to the protein solution (in 100 mM Tris pH 8.0, 100 mM NaCl) and incubating at 4 °C overnight. Unreacted spin-label was removed by size-exclusion chromatography. Completion of the spin-labeling reaction and attachment of a single spin-label on the protein was confirmed by LC/MS. Control spin-labeling reactions using wild-type UP1 protein showed that the native cysteines at positions 43 and 175 are inaccessible. PRE data were recorded and analyzed as described previously[26,54].

Residual dipolar couplings (RDCs) for $^1$H$^N$-$^{15}$N bonds were measured on a ~200 μM sample of the UP1/12-mer RNA complex aligned in Pf1 phage (~10 mg/mL) using the IPAP experiment[55]. RDCs were best-fitted to the structure by singular value decomposition (SVD) using PALES[56].

**X-ray crystallography**. Screening for crystallization conditions was done using commercial screens (QIAGEN and Hampton Research) at 20 °C. UP1 (in 20 mM HEPES pH 7.0, 100 mM NaCl, and 1 mM TCEP) was mixed with 12-mer RNA in a 1:1 molar ratio (0.95 mM protein–RNA complex concentration) and incubated on ice for 1 h. Crystallization drops were set-up by mixing 100 nL of complex and 100 nL of reservoir using the sitting drop vapor diffusion method. Crystals with hexagonal plate morphology were obtained in 0.2 M sodium citrate and 20% PEG 3350 after one week. The crystals were cryoprotected in reservoir solution supplemented with 20% (v/v) ethylene glycol and flash frozen in liquid nitrogen. X-ray diffraction data were collected at the European Synchrotron Radiation Facility (ESRF). All diffraction images were processed by XDS[57]. The CCP4 package[58] was used for all subsequent data analysis. The structure of the UP1/12-mer RNA complex was solved by molecular replacement with the program Phaser[59] using the coordinates of UP1 bound to modified telomeric DNA (PDB code: 1U1R)[60] as search model. Model building was performed manually in Coot[61] and refinement was done using Phenix[62].

**Restrained molecular dynamics simulations**. MD simulations were performed using GROMACS[63] and PLUMED[64]. The systems were prepared using the AMBER99SB-ILDN force field[65] in tandem with the TIP3P water model[66]. After solvation in a cubic box (100 Å side including 21,300 water molecules and 250 Å side including 531,000 water molecules for the UP1/12-mer RNA and the UP1/pri-mir-18a complex, respectively), a short initial energy minimization of 100 steps was performed to resolve steric clashes between solvent atoms and the complex. Structures were refined using simulated annealing simulations. Bonds were constrained using LINCS[67] and water was kept rigid using SETTLE.The temperature was adjusted with a period of 20 ps between 300 K and 100 K. The SAXS data set of the UP1/12-mer RNA complex encompassed data points with scattering wavenumbers between 0.025 Å$^{-1}$ and 0.685 Å$^{-1}$. Intensities up to 0.5 Å$^{-1}$ were fitted with a polynomial of 16$^{th}$ degree. From this fit, 43 intensities were calculated for wavenumbers between 0.03 and 0.45 Å$^{-1}$ in steps of 0.01 Å$^{-1}$. These representative intensities were utilized as restraints. For the UP1/pri-mir-18a complex representative intensities between 0.08 Å$^{-1}$ and 0.22 Å$^{-1}$ were calculated and used as restraints as described above. An approximate scaling factor relating calculated and measured SAXS intensities was estimated by the average ratio between the experimental SAXS intensities as taken from the fit and the first round of calculated SAXS intensities. SAXS intensities were calculated using the Debye formula and standard atomistic structure factors corrected for the effect of the solvent[68].

Metainference[69] with a Gaussian likelihood per data point on the representative SAXS intensities was applied every 10th step. Note, that metainference in the approximation of the absence of dynamics as used here (without replicas and a standard error of the mean set to zero) is equivalent to the Inferential Structure Determination approach[70]. The attributes of the uncertainty parameter were initially set to large values to allow a slow increase of the restrain force. Additionally, an additional scaling factor between the experimental data was sampled using a flat prior between 0.9 and 1.1.

The protein–RNA interface was restrained by harmonic upper-wall potentials centered at 3.5 Å applied on the distances between the centers of the respective rings (Phe17–A4, Phe59–G5, Phe108–A9, and Phe150–G10) with energy constants of 1000 kJ/mol. Furthermore, two crystallographic salt bridges (Arg75–Asp155 and Arg88–Asp157, distance between side-chain nitrogens and oxygens restrained to be below 5 Å) between the two RRM domains were restrained with similar potentials centered at a distance of 4 Å applied on the distances between their charged groups. Secondary structures identified by STRIDE[71] from the crystal structure were restrained using an upper-wall potential on the rmsd of the backbone atoms of residues involved with an energy constant of 10000 kJ/mol, centered at 0 Å.

For the UP1/12-mer complex RDCs restraints were applied using the θ-method[72]. To take into account for the multiple possible alignments of the molecule with the phage, RDCs were calculated as averages over two replicas and a linear restraint with a slope of −20,000 kJ/mol was applied on the correlation between the average and experimental RDCs. Each replica was independently restrained with all the formerly introduced restraints.

After 300 preliminary annealing cycles (for a nominal simulation time of 6 ns per refinement), the refined structure was chosen as the one with the lowest

metainference energy among those sampled at 300 K in the latest 30 cycles. The quality of the structure was then further assessed using PROCHECK[73] and the SAXS/SANS profiles and RDCs were independently confirmed using CRYSOL[74] and PALES[56], respectively.

**Isothermal titration calorimetry**. ITC (isothermal titration calorimetry) measurements were carried out at 25 °C using an iTC200 calorimeter (GE Healthcare). Both protein and ligand were exchanged into NMR buffer without DTT. Protein concentrations in the range of 300–1000 μM, depending on the affinity of the interaction, were injected into the sample cell containing RNA with a concentration of 20–100 μM. Titrations consisted of 20 injections of 2 μL or 26 injections of 1.5 μL with a 3-min spacing between each injection. After correction for heat of dilution, data were fitted to a one-site binding model using the Microcal Origin 7.0 software. Each measurement was repeated at least three times.

**Static light scattering**. Static light scattering (SLS) experiments were performed on a S75 10/300 size-exclusion column (GE Healthcare) connected to a Viscotek Tetra Detector Array (TDA) instrument equipped with refractive index (RI), light scattering, viscosity and photo diode array (PDA) detectors (Malvern Instruments). Sample volume of 100 μL was injected onto the column pre-equilibrated with NMR buffer and the flow rate was set to 0.5 mL/min. Calibration was done with BSA (bovine serum albumin) at a concentration of 4–5 mg/mL. Data were analyzed by the OmniSEC software using refractive index increment ($dn/dc$) values of 0.185 mL/g and 0.17 mL/g for protein and RNA samples, respectively[75].

**Small angle X-ray/neutron scattering**. SAXS data of UP1, pri-mir-18a, and UP1/pri-mir-18a complex were recorded at the X33 beamline of the European Molecular Biology Laboratory (EMBL) at Deutsches Elektronen Synchrotron (DESY, Hamburg) at 15 °C. The scattering curves were measured with 120-s exposure times (8 frames, 15 s each) for concentrations in the range of 1–10 mg/mL. The scattering intensity was measured covering the momentum transfer range $0.007 < q < 0.63 \text{ Å}^{-1}$. Individual frames collected during the exposure time were compared to check for radiation damage before averaging. Scattering of buffer measured before and after each sample was averaged and subtracted from the scattering of the sample. SAXS data of the UP1/12-mer complex were recorded on a BioSAXS-1000 instrument (Rigaku) equipped with a Pilatus detector using 20 μL of sample in a capillary tube. All SAXS data were processed using the ATSAS software package[76]. The radius of gyration ($R_g$) and the maximum dimension ($D_{max}$) values were obtained from the GNOM program, which evaluates the pair-distance distribution function, P($r$)[77]. For ab initio modeling of the protein–RNA complex, three scattering profiles corresponding to individual components and the complex were used as input for the multiphase modeling program MONSA[78]. Theoretical scattering curves were calculated using CRYSOL[74].

SANS data were recorded at the large dynamic range diffractometer D22 at the Institut Laue-Langevin (ILL) Grenoble, France, using a neutron wavelength of 6 Å and a detector-collimator set-up of 2 m/2 m. The scattering intensity was measured covering the momentum transfer range $0.02 < q < 0.35 \text{ Å}^{-1}$. Sample and buffer volumes were 200 μL and exposed for 60 min. The 2D detector signals of the samples were corrected for detector efficiency, empty cell scattering, directly calibrated (against water), and azimuthally averaged using ILL in-house software (Gosh, R. E., Egelhaaf, S. U. and Rennie, A. R. A Computing Guide for Small Angle Scattering Experiments. Technical Report ILL06GH05T, 2006, Institut Laue-Langevin, Grenoble, France). The final 1D scattering curves were further analyzed using the ATSAS software.

**SHAPE analysis**. Pri-mir-17-18a-19a (or pri-mir-18a) in complex with purified hnRNP A1 or UP1 proteins (125 nM) were assembled in folding buffer (100 mM HEPES pH 8, 6 mM $MgCl_2$) using 160 nM RNA prior to treatment with $N$-methylisatoic anhydride NMIA (Invitrogen), as the modifying agent, as recently described[79]. RNA was phenol-extracted and ethanol-precipitated and then subjected to primer extension analysis. For primer extension, equal amounts of NMIA-treated and untreated RNAs (10 μL) were incubated with 0.5 μL of antisense 5′-end $^{32}$P-labeled primers (SHAPE1-3; Supplementary Table 3), which were used for the analysis of regions comprising miR-17, miR-18a, and miR-19a, respectively. Primer extension and SHAPE data processing was conducted, as previously described[80].

**Data availability**. Atomic coordinates and structure files for the UP1/12-mer RNA crystal structure have been deposited in the Protein Data Bank (http://www.pdb.org/) with accession code 6DCL. Coordinates for the structural models of the 1:1 complexes of UP1/12-mer and UP1/pri-mir-18a are available upon request. Uncropped gels are presented in Supplementary Fig. 7. Other data are available from the corresponding authors upon reasonable request.

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

## Acknowledgements

We thank Marianne Keith (MRC HGU, University of Edinburgh) for Immuno-fluorescence data, Robert Janowski for collecting the X-ray diffraction data, and Ralf Stehle for help with in-house SAXS measurements. We are grateful to the beamline staff at DESY (Hamburg) and Anne Martel at ILL (Grenoble) for support with SAXS and SANS measurements, respectively. We acknowledge the use of the X-ray crystallography platform at the Helmholtz Zentrum München, NMR access at the Bavarian NMR Center and SAXS measurements at the facility of the SFB1035 at Department Chemie, Technical University of Munich. This work was supported by the Deutsche For-schungsgemeinschaft through grants SFB1035 and GRK1721 (to M.S.), MRC core funding and the Wellcome Trust (Grant 095518/Z/11/Z to J.F.C.), Medical Research Council Career Development Award (G10000564 to G.M.). Wellcome Trust Center Core Grants (077707 and 092076 to G.M.). M.M., A.J., and C.C. acknowledge the support of the Technical University of Munich—Institute for Advanced Study, funded by the German Excellence Initiative and the European Union Seventh Framework Programme under grant agreement n° 291763.

## Author contributions

H.K. performed molecular biology, protein/RNA preparation, ITC, NMR, crystal-lographic structure determination, RNA modeling, and SAXS analysis; N.R.C. and G.M. performed footprinting and miRNA processing assays both in vitro and in cells in culture, N.F. performed SHAPE experiments, N.F. and A.N.J. analyzed SHAPE data, B.S., M.M., A.J., and C.C. performed structural modeling and restrained molecular dynamics, F.G. analyzed SAXS and SANS data, A.D. performed NMR of RNA samples. H.K., G.M., J.F.C., and M.S. conceived and designed the project. H.K., G.M., J.F.C., and M.S. wrote the paper. All authors discussed the results and commented on the manuscript.

## Additional information

**Competing interests:** The authors declare no competing interests.

