## [Peer Review File · Nature Communications]

Reviewers' comments:

Reviewer #1 (Remarks to the Author):

Summary:

The authors present a structure of hnRNP A1 bound to the terminal loop of miR-18a, suggesting a specific interaction between the protein and the RNA. They also show that the binding event leads to a change in the RNA conformation, specifically leading to slight melting near the terminal loop. The authors suggest that binding of the terminal loop by hnRNP A1 somehow leads to structural changes near the base of the stem, which somehow activates Drosha processing.

Comments:

Given that several structures of hnRNP A1 are available, especially with DNA or RNA, the structural data that the manuscript only provides incremental information. The proposed mechanism is interesting. However, although the NMR data shows that there is a slight opening of the loop with hnRNP A1 binding, more experiments should be done to support the mechanism to explain the real functional outcome—Drosha processing. Here are some suggestions to further test the authors' proposed hypothesis.

1. For structural data, it would be helpful to use figures to show how your structures are different from previously published ones.
2. All the experiments are performed with hnRNP A1 and pri-miRs. Since hnRNP A1 supposedly enhances Drosha/DGCR8 activity, it would make sense to do the RNA binding/structure experiments in the presence of them. DGCR8 is supposed to bind the terminal loop where hnRNP A1 binds. So do they bind at the same time? Are the RNA structural changes the same in both contexts? What is the affinity of hnRNP A1 for the loop, and would it compete against DGCR8?
3. In Fig 5, what happens if you use mutant UP1 or mutant hnRNP A1? Do you not observe the changes near the basal stem?
4. In Fig 6, how does the processing change when pri-miR-18a is processed on its own rather than in the context of the cluster? If the mechanism is traveling down the stem, the neighboring pri-miRs shouldn't matter. Yet, in a previous paper from the same group (Guil et al, 2007), they have shown that the context does matter.
5. In Fig 6, RNA mutations are confounding because they probably affect both hnRNP A1 binding and Drosha binding/processing. Some of these are drastic mutations, and Drosha processing has been shown to be sensitive to mutations, given the motifs that multiple groups identified. Can you get these results with hnRNP A1 mutations? Every RNA mutation should also have controls with Drosha but without hnRNP A1, so that hnRNP A1 effects are clear.
6. The mechanism of opening up the RNA down the stem is difficult to imagine. Does your SHAPE data agree with this mechanism?
7. Fig 1C: How do the pri levels compare to the mature levels?

Reviewer #2 (Remarks to the Author):

The authors of Manuscript "Structural basis for terminal loop recognition and processing of pri-miRNA-18a by hnRNP A1" by Kooshapur et al. use an integrated structural biology approach to reveal the structural basis of how auxiliary protein factor hnRNP A1 positively regulates the processing of miRNA-18a primary transcript. The authors have methodically and very artfully demonstrated 1. UP1-M9 nuclear localization is essential for the miRNA processing using in vivo processing assays; 2. Specific contacts/interactions between UP1 and pri-miRNA-18a using EMSA and a number of mutants and analysis of NMR data; 3. Proposed a monomeric structural model of the UP1-12mer complex based on NMR and crystallographic data; 4. Validated the structural

model using a number of both UP1 and the RNA mutants; 5. Further illustrated destabilized the terminal loop and the adjacent stem using NMR and footprinting experiments; subsequent allosteric effect on the stability of internal and terminal duplex using NMR, SHAPE and a number of carefully designed RNA mutants. Both of the effects on the stability of the RNA are the consequences of UP1 binding at the terminal loop; 6. Further validated their structural model using SAXS and NMR data. The authors proposed a mechanistic model of how hnRNP A stimulates the processing of pri-miRNA-18a. In conclusion, the authors' evidence is compelling and the work represents a major step toward understanding of the miRNA biogenesis.

Minor suggestions/comments

1. Change the last sentence in the abstract from "Our results highlightas a general principle of miRNA biogenesis and regulation" to "Our results highlightas potentially a general ...", since in this paper the authors demonstrated only one case study, which cannot logically be extrapolated to a general principle.

2. The biggest weakness of this study is the monomeric structural model, which was constructed based on the dimeric crystal structure and validated by sparse NMR data. Given the nanomolar affinity and relatively a small size of the complex (UP1-12mer), it seems so obvious that the structure of the UP1-12mer complex is attainable using NMR. The authors might want to discuss/reveal why this is not the case.

I suggest to rephrase the sentence "Although the stoichiometry ..., the RNA contacts are expected to be conserved,...." (pg 8). The UP1-12mer model proposed by the authors is certainly consistent with their experimental results such as footprinting, SHAPE etc. It is a reasonable model. However, there is no direct evidence to suggest in detail that the structural contacts between the RNA and the protein is conserved between the monomeric solution structure and the dimeric crystal structure, as the authors haven't really determined (or presented) the solution structure. In order to conserve all contacts between the dimeric structure and the monomeric model, one of the RRM2s in the crystal structure have to rotate by 180° and translate about half of the unit cell length with all RNA-protein contact unchanged to replace another RRM2, while the 3-residue linker of the 12mer RNA has to undergo a very sharp torsion angle rotation to accommodate such a change. The authors could convince themselves that their statement is too much an extrapolation simply by calculating the differences between torsion angles of the RNAs in the structure and the model.

3. Somewhat related to the comment above, I suggest the authors provide another figure alongside of the Fig. 3e to show the plot of RDC cal. based on crystal structure of one of monomers vs. exp. RDC (label residues of outlier in both the protein and RNA). Such a plot would be more indicative of how close the structure and the model are.

4. In pg 9 the authors described how interactions between hydrophobic residues phe in UP1 and ribose rings in RNA contribute to the interaction/binding. Not sure what type of interactions, hydrophobic/hydrophilic/ionic/van der Waal the authors refer to. In my view, those contacts contribute nothing to the interaction/binding/recognition.

5. Did authors look into the crystal contacts between the two UP1 molecules to see if there is any salt bridges/H-bond that contribute to the dimeric formation? If any, further exploring a possibility of mutations to remove such interactions might possibly result in monomeric crystals.

6. It would be interesting to show the EMSA with the 12mer, 17mer and pri-miRNA-18a in Fig. 1d.

7. The ITC result for the UP1+17mer RNA titration is mysterious. How many trials were performed? Your discussion on the NMR result (the last paragraph in pg 12) almost suggests the interaction between UP1 and the 12mer/17mer is identical, contradicting the ITC result. Please elaborate on this.

8. Change Fig. 4b to Fig. 4c in the second paragraph pg 13.

9. Label the peak on the right side of G52 in Fig. 4c. If not assigned, please indicate so.

10. Label domains with RRM1 and 2 in Fig. 5a.

11. Explain why the B-factors (Table 2) are so high.

12. Provide the ratio plot, para/dia vs. residue number in Fig. 4b in Supplementary Info. This plot would be more indicative of what have changed upon binding.

13. Plot pairwise distance rmsd between the crystal structure and the model along Fig. 4d in

Supplementary Info. Change label "crystal structure/solution structure" to "crystal structure/solution model"

Reviewer #3 (Remarks to the Author):

This is a comprehensive study of processing of micro-RNA by a nuclear RNA binding protein, hnRNP A1, which is implicated in a diversity of RNA processing systems. Using several structural and biophysical approaches the authors put together a picture that sheds light on the mechanism underpinning hnRNP A1 processing of a particular miRNA, of general interest for miRNA processing. One of the approaches used involves restrained molecular dynamics simulation, in which experimental data from Small Angle X-ray Scattering and Residual Dipolar Coupling NMR were used to guide the simulations towards a structural model in agreement with the observation. This fits well in the overall integrative approach of the manuscript. The molecular dynamics part of the methods section need some further clarifications and explanations, as indicated below.

The length of the production simulations (ie, the amount of sampling), and the size of the system (dimensions of the PBC box) should be stated.

References are lacking for the AMBER99SB-ILDN forcefield, the TIP3P water model, and the SVD used for confirmation of the RDCs.

TIP3P is a rigid water model - how was this maintained in the simulations?

The unit (kJ/mol) for the force constants is an energy-unit, not a force constant.

How is the distance between the charged groups of the salt-bridges defined?

Point-by-point response

Structural basis for terminal loop recognition and processing of pri-miRNA-18a by hnRNP A1

Kooshapur et al

New data and figures added

We have performed additional experiments that we believe clarify issues raised by Reviewer 1 (Points 4, 5, and 6):

- We are now including **two entirely new figures**, related to functional assays of miRNA processing in cells (**Supplementary Figure 6** and **Figure 7**) that further support our conclusions and clarify the points raised.
- Of particular interest, we constructed a set of pri-mir-16 chimeras that harbor wild type of mutated pri-mir-18a terminal loops in order to be able to evaluate how mutations in defined regions of pri-mir-18a affect its processing in living cells (**Figure 7**).
- We have analyzed the secondary structure of pri-mir-18a based on the SHAPE and NMR data and assessed the changes in SHAPE reactivity upon binding of UP1 and hnRNP A1 (**new Figure 5d**), which provides further support for our conclusions.

Point-by point response to reviewers

Reviewer #1 (Remarks to the Author):

Summary:

The authors present a structure of hnRNP A1 bound to the terminal loop of miR-18a, suggesting a specific interaction between the protein and the RNA. They also show that the binding event leads to a change in the RNA conformation, specifically leading to slight melting near the terminal loop. The authors suggest that binding of the terminal loop by hnRNP A1 somehow leads to structural changes near the base of the stem, which somehow activates Drosha processing.

Comments:

Given that several structures of hnRNP A1 are available, especially with DNA or RNA, the structural data that the manuscript only provides incremental information.

We respectfully disagree. In fact we believe that the structural data are by any means not incremental but rather provide the first experimental evidence and structural model for a 1:1 complex of UP1 with RNA. Our crystal structure of the UP1/12-mer RNA complex solved at 2.5 Å resolution is novel (Table 2; Fig. 3a; Supplementary Fig. 4a), although we would agree that the 2:2 stoichiometry seen in the crystal with two molecules of UP1 and two RNA chains in the asymmetric unit (Supplementary Fig. 4a), is similar to the previously reported structure of UP1 with single-stranded telomeric DNA (Ref. 24, Ding et al. (1999) PMID:10323862). Notably, we unambiguously show for the first time that the peculiar 2:2 stoichiometry does not represent the UP1:RNA complex in solution, and provide for the first time a structural model for a 1:1 UP1:RNA complex that represents the solution conformation. So far published structures of UP1 complexes with DNA and RNA (including recent work by the Ma group (*Nature Commun* 2018)) have not addressed the arrangements of the tandem

domains in solution and focus on presenting structural information for the protein-RNA interface for the individual domains. The Allain lab (*eLife* 2017) has proposed a 1:1 model of UP1 in the context of splicing regulation where the topology of the protein-RNA interface is distinct from what we report for the interaction with pri-miRNA. Moreover, the destabilization or partial “unwinding” of an RNA hairpin by UP1 (Unwinding Protein 1) has not been addressed by previous literature. Thus, we are convinced that our work provides truly novel insight into an unprecedented mode of RNA recognition by hnRNP A1.

The proposed mechanism is interesting. However, although the NMR data shows that there is a slight opening of the loop with hnRNP A1 binding, more experiments should be done to support the mechanism to explain the real functional outcome—Drosha processing. Here are some suggestions to further test the authors’ proposed hypothesis.

1. For structural data, it would be helpful to use figures to shown how your structures are different from previously published ones.

The relative domain arrangement of the two RRM domains is conserved with only minor variations seen in structures of free or bound UP1. This is shown in the Supplementary Figure 4e, where we show a superposition of UP1 free (NMR structure PDB: 2LYV) (Barraud et al 2013), UP1 bound to three nucleotides (crystal structure PDB: 4YOE) (Morgan et al 2015) and UP1 bound to DNA (crystal structure PDB:1U1Q) Myers et al 2004. However, as discussed in the comment above, the stoichiometry of the protein-RNA complex in solution has not been experimentally demonstrated with the exception of a recent model proposed by the Allain lab in the context of splicing regulation (*eLife* 2017). A structure from the Tolbert lab shows only three nucleotides bound, and thus does not provide information on RNA recognition involving both domains, while the domain arrangement is the canonical one observed in solution free or bound.

2. All the experiments are performed with hnRNP A1 and pri-miRs. Since hnRNP A1 supposedly enhances Drosha/DGCR8 activity, it would make sense to do the RNA binding/structure experiments in the presence of them. DGCR8 is supposed to bind the terminal loop where hnRNP A1 binds. So do they bind at the same time? Are the RNA structural changes the same in both contexts? What is the affinity of hnRNP A1 for the loop, and would it compete against DGCR8?

We appreciate these comments and suggestions and agree that these are important questions. Note, that Drosha and DGCR8 are part of the microprocessor complex that is highly challenging to reconstitute *in vitro* for structural studies. The model of Drosha/DGCR8 binding to pri-miRNA, presented in Kwon SC et al. 2016 Cell, proposes that the RNA-binding heme domains (Rhed) in DGCR8 extends towards the terminal loop but do not cover it. This should leave room for the binding of auxiliary factors, such as hnRNP A1 or Lin28 to pri-miRNA terminal loops. We do agree that the question posed by the reviewer is highly relevant but to recapitulate all these experiments using recombinant Drosha and/or DGCR8 is a difficult task and we certainly feel that this is beyond the scope of this manuscript.

3. In Fig 5, what happens if you use mutant UP1 or mutant hnRNP A1? Do you not observe the changes near the basal stem?

It is not entirely clear to which mutant UP1 forms does this Reviewer refer to. We have extensively used UP1 mutant proteins (in the context of UP1-M9 fusions in cells in culture) to show that both RRMs are required to bind RNA and to promoter miRNA processing. These experiments also revealed the contribution of specific residues within the RNP-1 submotifs. Thus, use of these mutants in the probing experiments will not be very informative, as they will not bind the terminal loop of pri-mir-18a.

4. In Fig 6, how does the processing change when pri-miR-18a is processed on its own rather than in the context of the cluster? If the mechanism is traveling down the stem, the

neighboring pri-miRs shouldn't matter. Yet, in a previous paper from the same group (Guil et al, 2007), they have shown that the context does matter.

We have now done the *in vitro* processing of pri-miR-18a on its own (**Supplementary Figure 6**). Our new results recapitulate the findings from the pri-miR-17-18a-19 cluster presented already. Now, it is even more evident that the pri-miR-18a[U21A/U29A/U34A], which displays weak hnRNP A1 binding affinity in RNA pull down experiments, is processed much less efficiently by Drosha as compared to the wild-type controls. Importantly, the pri-miR-18a[5GC_internal] mutant displayed strong hnRNP A1 binding but no Drosha-mediated processing.

5. In Fig 6, RNA mutations are confounding because they probably affect both hnRNP A1 binding and Drosha binding/processing. Some of these are drastic mutations, and Drosha processing has been shown to be sensitive to mutations, given the motifs that multiple groups identified. Can you get these results with hnRNP A1 mutations? Every RNA mutation should also have controls with Drosha but without hnRNP A1, so that hnRNP A1 effects are clear.

This reviewer raises a very good point.

In our previous publication (Guil and Caceres, 2007, PMID: 17558416) we clearly demonstrated that in the absence of hnRNP A1 Drosha does not cleave pri-miR-18a. To address the reviewer's concerns about RNA mutations influencing Drosha binding/processing we have now generated a set of pri-miR-16 chimeras that harbor wild type or mutated pri-mir-18a terminal loops (**Figure 7**). Importantly, we observe that Drosha efficiently processes all but one of the pri-miR-16/18a chimeras, with the processing of pri-miR-16/18a[5GC] being reduced. The reduced processing of this chimeric construct harboring 5GC in the vicinity of the terminal loop shows that conformational changes to apical part of the pri-miR stem can directly affect Microprocessor activity. This supports the reviewer's concerns that some mutations to pri-miR terminal loop can directly affect Drosha processing. Importantly, the processing of pri-miR-16/18a[5GC_internal] within the context of the pri-mir-16-18a chimera was efficient (**Figure 7**), unlike the processing of pri-miR-18a[5GC_internal], which was abolished (**Fig. 6b**). These results strongly support our hypothesis that unwinding of pri-mir-18a by hnRNP A1 can spread from the terminal loop towards the stem and is essential for stimulation of miR-18a biogenesis.

6. The mechanism of opening up the RNA down the stem is difficult to imagine. Does your SHAPE data agree with this mechanism?

Please note, that we propose that UP1 binding destabilizes the RNA stem and renders it more dynamic, but does not lead to a complete melting except in the region of the UP1 binding site. Indeed the enzymatic probing (**Figure 5b**) and SHAPE data support the destabilization of the stem. We have further analyzed the SHAPE data and used this together with our NMR evidence for base-pairing in the RNA to generate secondary structures of the pri-mir-18a free and when bound to UP1. These data are shown in a **new panel in Figure 5d** based on the SHAPE data shown in **Supplementary Figure 5** and are discussed in a new section in the results part.

Clearly, UP1 binding shows significantly increased SHAPE reactivity in the loop proximal region of the stem, consistent with the proposed destabilization.

7. Fig 1C: How do the pri levels compare to the mature levels?

We have tried measuring the pri-miR-18a by qRT-PCR (with several primer sets) but its levels were undetectable in our hands. This is perhaps not surprising as Drosha efficiently processes all other pri-miRs that surround pri-miR-18a (pri-miR-17, miR-miR-19 and pri-miR-92) in the presence or absence of hnRNP A1. It is very common not to see pri-miRs of efficiently processed miRs.

Reviewer #2 (Remarks to the Author):

The authors of Manuscript “Structural basis for terminal loop recognition and processing of pri-miRNA-18a by hnRNP A1” by Kooshapur et al. use an integrated structural biology approach to reveal the structural basis of how auxiliary protein factor hnRNP A1 positively regulates the processing of miRNA-18a primary transcript. The authors have methodically and very artfully demonstrated 1. UP1-M9 nuclear localization is essential for the miRNA processing using in vivo processing assays; 2. Specific contacts/interactions between UP1 and pri-miRNA-18a using EMSA and a number of mutants and analysis of NMR data; 3. Proposed a monomeric structural model of the UP1-12mer complex based on NMR and crystallographic data; 4. Validated the structural model using a number of both UP1 and the RNA mutants; 5. Further illustrated destabilized the terminal loop and the adjacent stem using NMR and footprinting experiments; subsequent allosteric effect on the stability of internal and terminal duplex using NMR, SHAPE and a number of carefully designed RNA mutants. Both of the effects on the stability of the RNA are the consequences of UP1 binding at the terminal loop; 6. Further validated their structural model using SAXS and NMR data.

*The authors proposed a mechanistic model of how hnRNP A stimulates the processing of pri-miRNA-18a. In conclusion, the **authors’ evidence is compelling and the work represents a major step toward understanding of the miRNA biogenesis.***

We are thankful to this reviewer for these positive comments.

Minor suggestions/comments

1. Change the last sentence in the abstract from “Our results highlightas a general principle of miRNA biogenesis and regulation” to “Our results highlightas potentially a general ...”, since in this paper the authors demonstrated only one case study, which cannot logically be extrapolated to a general principle.

We have followed this suggestion and have modified the text accordingly.

2. The biggest weakness of this study is the monomeric structural model, which was constructed based on the dimeric crystal structure and validated by sparse NMR data. Given the nanomolar affinity and relatively a small size of the complex (UP1-12mer), it seems so obvious that the structure of the UP1-12mer complex is attainable using NMR. The authors might want to discuss/reveal why this is not the case.

We agree that in principle the molecular weight of the complex is clearly within the range of NMR structure determination and in fact have put very much effort in being able to apply this. However, we faced a number of challenges that render an NOE-based structural analysis difficult. There is significant line-broadening in the 12-mer RNA-bound spectra, which prevented us from obtaining sufficient intermolecular NOEs to define the protein-RNA interface and also domain-domain interface. In spite of substantial efforts we could not find conditions to overcome these problems. The line-broadening likely reflects the presence of some conformational dynamics in the protein-RNA complex that may affect the domain-domain interface and the linker connecting the two domains.

On the other hand NMR chemical shift perturbation are fully consistent with the overall RNA interface seen in the crystal structure, and demonstrate that both domains are involved in the binding interface. Therefore, we performed a semi-rigid structure calculation to determine a structural model for the 1:1 complexes. We would like to stress that this model is consistent with experimental NMR restraints measured on the 1:1 complex, i.e. RDC and PRE data that report on domain orientation and arrangements, respectively. In addition, SAXS data were used to define the 1:1 model, which is also fully consistent with our biochemical and biophysical data.

We agree that our integrative approach relies on sparse data and therefore refer to it as a structural model. We have adapted the text to check that this is stated appropriately. In any case, we believe that it is important to present the 1:1 model, and stress its significance in

representing the solution conformation of the UP1-RNA complex, as numerous crystallographic studies also in recent years have, in our view, overlooked this important aspect.

I suggest to rephrase the sentence "Although the stoichiometry ..., the RNA contacts are expected to be conserved,..." (pg 8). The UP1-12mer model proposed by the authors is certainly consistent with their experimental results such as footprinting, SHAPE etc. It is a reasonable model. However, there is no direct evidence to suggest in detail that the structural contacts between the RNA and the protein is conserved between the monomeric solution structure and the dimeric crystal structure, as the authors haven't really determined (or presented) the solution structure.

*In order to conserve all contacts between the dimeric structure and the monomeric model, one of the RRM2s in the crystal structure have to rotate by 180° and translate about half of the unit cell length with all RNA-protein contact unchanged to replace another RRM2, while the 3-residue linker of the 12mer RNA has to undergo a very sharp torsion angle rotation to accommodate such a change. The authors could convince themselves that their statement is too much an extrapolation simply by calculating the differences between torsion angles of the RNAs in the structure and **the model**.*

We thank the reviewer for these important comments and clarify them as follows:

(1) We agree with this reviewer that we do not have high-resolution atomic details that prove that the RNA recognition is conserved between the crystal structure and the solution model. This is likely the case for the flanking regions of the central AG motifs in the RNA. We have rephrased the statement and added some cautionary remarks in the text. Nevertheless, our NMR chemical shift perturbations and ITC data clearly show that the canonical RNA binding surfaces of both RRMs are involved in solution as seen in the crystal structure, and this is what the statement intended to say.

(2) The structural rearrangement suggested by the reviewer to go from the crystal structure to the solution conformation is not required at all. To the contrary, for the protein structure there is basically no rearrangement involved and the RRM1-RRM2 domain arrangement is the same as observed in all solution structures published and present also in crystallographic structures of UP1, as can be seen in Suppl. Fig. 4e,f. On the other hand, the RNA conformation clearly has to adapt compared to the crystal structure. To "morph" the RNA from the 2:2 crystal structure to the 1:1 solution conformation, it is easiest to envision cutting the RNA strands in the region between the RRM domains and reconnecting the different halves.

3. Somewhat related to the comment above, I suggest the authors provide another figure alongside of the Fig. 3e to show the plot of RDC cal. based on crystal structure of one of monomers vs. exp. RDC (label residues of outlier in both the protein and RNA). Such a plot would be more indicative of how close the structure and the model are.

Thank you for this suggestion. In fact we had done this but had not included the data in the previous version of the manuscript. To validate that the presence of the typical RRM1-RRM2 domain arrangement (which is seen in all structures reported so far) in our 1:1 structural model of the UP1-12-mer complex we have correlated experimental NMR RDC data with the crystal structure and our model. The RDC Q-factor for UP1 (molecule A) in the crystal structure is 0.34, while a chimera assembled from RRM1 of molecule A and RRM2 of molecule B is 0.47. This clearly supports that in solution the UP1-RNA complex adopts the "canonical" RRM1-RRM2 arrangement that is also present for the two protomers in the crystal structure. The RDC plots are shown in the new **Supplementary Fig. 4b**.

Note that the UP1 tandem domains in the crystal structure and in the 1:1 solution model overlay with an rmsd of 2.8 Å as shown in **Suppl. Fig. 4e**.

4. In pg 9 the authors described how interactions between hydrophobic residues phe in UP1 and ribose rings in RNA contribute to the interaction/binding. Not sure what type of interactions, hydrophobic/hydrophilic/ionic/van der Waal the authors refer to. In my view, those contacts contribute nothing to the interaction/binding/recognition.

We have removed the two sentences.

5. Did authors look into the crystal contacts between the two UP1 molecules to see if there is any salt bridges/H-bond that contribute to the dimeric formation? If any, further exploring a possibility of mutations to remove such interactions might possibly result in monomeric crystals.

We have analyzed the various dimer interfaces in the crystal structure and described the intermolecular contacts in the dimer in the legend of Supplementary Figure 4.

“ the two UP1 molecules form a dimer mediated by intermolecular contacts involving Glu11 in $\alpha 0$, Asp94 in the inter-RRM linker and four residues from RRM2 (Ile164, Lys166, Tyr167 and His173). The contacts at the dimer interface are the same as described for the UP1-DNA complex (Ding et al. 1999) and include hydrogen bonding (Glu11-His173 and Tyr167-Ile164) and a salt bridge between Asp94 and Lys166.

It seems that the interfaces in between different protomers are less strong but exhibit a combination of charged and hydrophobic interactions. We have not yet attempted to modify these interfaces to promote crystallization of a monomeric complex but consider this an excellent suggestion for future work.

6. It would be interesting to show the EMSA with the 12mer, 17mer and pri-miRNA-18a in Fig. 1d.

We agree that it is important to compare the different binding affinities. We have chosen to use ITC experiments for this as they provide more precise quantitative numbers and are performed free in solution. The data are shown in Table 1.

7. The ITC result for the UP1+17mer RNA titration is mysterious. How many trials were performed? Your discussion on the NMR result (the last paragraph in pg 12) almost suggests the interaction between UP1 and the 12mer/17mer is identical, contradicting the ITC result. Please elaborate on this.

We have repeated this measurement at least 3 times. The complex ITC binding profile of the UP1+17-mer interaction is possibly due to multiple-binding events and/or conformational changes associated with binding. In fact we believe that this unusual behavior is consistent with the relative poor stability of base pairing in the top part of the RNA stem as follows: Of the two UAG motifs in the 17-mer construct, the one located in the GUAGA pentaloop is directly accessible for interaction with UP1, whereas the guanosine in the second UAG motif is involved in base-pairing with a uridine forming the stem of the 17-mer (**Figure 4d,e**). Therefore, recognition of both UAG motifs requires unfolding of the 17-mer hairpin, which presumably causes the unusual ITC curve. It is noteworthy that, the additional G:C base-pair added at the beginning of the 17-mer RNA can, at least partially, stabilize the region in the vicinity of the second UAG motif. This artificial stabilization might explain the unusual ITC binding curve of the UP1-17-mer interaction. One could imagine that binding of UP1 to the first UAG motif induces melting of the hairpin in order to allow binding to the second UAG motif. Although we could not quantify the binding affinity of UP1 to the 17-mer RNA by ITC, the NMR spectra shown in **Supplementary Fig. 3c** indicate that binding to 12-mer and 17-mer are identical, as one would expect with both UAG motifs in the RNAs being recognized in a single-stranded conformation.

8. Change Fig. 4b to Fig. 4c in the second paragraph pg 13.

This has been corrected.

9. Label the peak on the right side of G52 in Fig. 4c. If not assigned, please indicate so.

We have now labeled this peak as not assigned on Fig. 4c.

10. Label domains with RRM1 and 2 in Fig. 5a.

This has been done.

11. Explain why the B-factors (Table 2) are so high.

There are two molecules (two protein and two RNA chains) in the asymmetric unit and the B-factors reported in Table 2 are overall average values for different chains. The second molecule in the asymmetric unit (chains B and D) has a significantly higher degree of disorder. Protein chains A and B have B-factors of 63 and 79 Å², respectively. RNA chains C and D have B-factors of 72 and 99 Å², respectively. Similar B-factors have been reported for structures at this resolution.

12. Provide the ratio plot, para/dia vs. residue number in Fig. 4b in Supplementary Info. This plot would be more indicative of what have changed upon binding.

We believe that the PRE data plotted within the same graphic as in Supplementary Fig. 4b give a convenient presentation to compare the data and prefer to leave the figure panel as is. The plot suggested by the reviewer also shows the overall similarities between the free and bound forms, but the information about the individual free and bound profiles is lost in this representation.

13. Plot pairwise distance rmsd between the crystal structure and the model along Fig. 4d in Supplementary Info. Change label “crystal structure/solution structure” to “crystal structure/solution model”

This has been done, as suggested.

Reviewer #3 (Remarks to the Author):

This is a **comprehensive study** of processing of micro-RNA by a nuclear RNA binding protein, hnRNP A1, which is implicated in a diversity of RNA processing systems. Using several structural and biophysical approaches the authors put together a picture that sheds light on the mechanism underpinning hRNP A1 processing of a particular miRNA, of general interest for miRNA processing.

One of the approaches used involves restrained molecular dynamics simulation, in which experimental data from Small Angle X-ray Scattering and Residual Dipolar Coupling NMR were used to guide the simulations towards a structural model in agreement with the observation. This fits well in the overall integrative approach of the manuscript.

The molecular dynamics part of the methods section need some further clarifications and explanations, as indicated below.

The length of the production simulations (ie, the amount of sampling), and the size of the system (dimensions of the PBC box) should be stated.

Details have now been added in the methods under “Restrained molecular dynamics”.
The box size was 100 Å and 250 Å for the 12-mer and the full length systems, respectively.

References are lacking for the AMBER99SB-ILDN forcefield, the TIP3P water model, and the SVD used for confirmation of the RDCs.

Citations have been added.

TIP3P is a rigid water model - how was this maintained in the simulations?

Citations and details have been added.

The unit (kJ/mol) for the force constants is an energy-unit, not a force constant.

This has now been corrected.

How is the distance between the charged groups of the salt-bridges defined?

The distance between side chain oxygens and nitrogens in the salt bridges has been restrained to be below 5 Å.

Reviewers' comments:

Reviewer #1 (Remarks to the Author):

The authors have previously made an interesting discovery that hnRNP A1 has a role in enhancing processing of pri-miR-18a. Current manuscript is about the mechanism of hnRNP A1. Two main parts to this work are 1) structural studies of hnRNP A1 with RNA to establish binding site and stoichiometry; and 2) the effects of hnRNP A1 on Drosha processing, by changing RNA structure. The first part is convincing, but many of the details are already known for hnRNP A1 interactions with RNA. And whether this is the only arrangement that would exist in the cell is still unclear since it seems rather dynamic and even the cluster itself might be dynamic. The second question on regulation is interesting, but the answer that the authors propose need more direct evidence. Here are some suggestions and comments.

1. From the NMR (Fig 4C), T1 footprinting (Fig 5B) and SHAPE data (Fig S5), the authors suggest a mechanism in which: (a) hnRNP A1 binding to the pri-18a terminal loop leads to partial opening/melting; and (b) destabilization of the pri-18a stem region. (a) The footprinting and SHAPE data show changes in signal when hnRNP A1 is added. However, there is less evidence for the statement "leads to partial opening/melting terminal loop" in the data presented. (b) The authors explain that there is stem destabilization hypothesis using data from NMR (Fig 4C), footprinting (5B) and SHAPE (data not shown? - Line 393- I thought this was not acceptable.). The current SHAPE data in Fig S5 does not clearly support the destabilization model since there is reduced SHAPE reactivity in the stem region when UP1 is present.

2. The resolution from NMR is not sufficient to derive high-resolution spatial information. And both footprinting and SHAPE data would benefit from having controls that do not completely change the RNA structure, such as mutant hnRNP A1. This was the original points #5 and #3. It is a simple question about whether a protein mutation in RRM leads to a change in the footprint or SHAPE or NMR. Moreover, whether a protein mutation in RRM leads to changes in processing. If the primary novelty that the authors are suggesting is the way UP1 sits on the terminal loop, this is a rather simple experiment and much more telling than elaborate chimeras where the RNA structures have changed a lot. The reviewer suggests that the authors look for changes in RNA when one of the RRMs is mutated. Since you know the complex structure, mutant proteins would be better as controls than in the absence of protein.

3. The 16/18 chimera experiments are rather indirect. As the authors showed, various RNA elements can have effects on processing, and why certain mutations they tested affect Drosha processing is still unclear. And it is unclear how one feature (such as the 5GC clamp) would interact with another. For example, how the pri-miR18a5GC interacts with hnRNP A1 might be different from the wild type miR-18a. Instead of controlling with miR16, it would be better to control with hnRNP A1 mutations.

4. ITC can be used to measure both affinity and stoichiometry. Can the authors use this instead of just the affinity values to determine the ratios?

5. Line 495 "How does binding of a regulator to the terminal loop of pri-miRNA affect Drosha...": Current model of primary microRNA processing includes that most of the terminal loop is critical for Drosha processing, even before any loop binding proteins are present. And it also suggests that the RNA motifs/structure are linked, thus making it difficult to assume that certain features are additive. Please refer to Nguyen et al. 2015, Kwon et al. 2015, Auyeong et al. 2013, Partin et al. 2017, and Roden et al., 2017. What the authors propose—that hnRNPA1 melts the loop and somehow propagates the changes to the Drosha cut site—is intriguing but requires more evidence. For example, the direct effect of hnRNP A1 needs to be shown, via reconstitution or using appropriate protein mutants. This is especially important since various structural and biochemical evidence suggest that a lot of the stemloop is already contacting Drosha and DGCR8. Moreover,

the manuscript would be stronger with a discussion that can reconcile the differences in the models.

Reviewer #2 (Remarks to the Author):

I would urge the authors to show the EMSA results as I recommended in the previous review for the following reasons:

1. The interpretation of ITC results depends on type of models one uses to fit the data, where the EMSA gives a straightforward binding profile without any assumption.
2. ITC does not always work, as authors have shown in UP1+17mer titration, even though NMR appears to indicate it should work and give an expected result.

I understand the EMSA gel pictures may look messy. But it will give readers a complete realistic and unhinged view about the complexes: whether they are in fast, slow or intermediate exchange regimes in addition to NMR results, which are usually measured at much higher concentration.

I am otherwise satisfied with all other responses from the authors.

Point-by-point response to reviewers' comments:

Reviewer #1 (Remarks to the Author):

The authors have previously made an interesting discovery that hnRNP A1 has a role in enhancing processing of pri-miR-18a. Current manuscript is about the mechanism of hnRNP A1. Two main parts to this work are 1) structural studies of hnRNP A1 with RNA to establish binding site and stoichiometry; and 2) the effects of hnRNP A1 on Drosha processing, by changing RNA structure. The first part is convincing, but many of the details are already known for hnRNP A1 interactions with RNA. And whether this is the only arrangement that would exist in the cell is still unclear since it seems rather dynamic and even the cluster itself might be dynamic. The second question on regulation is interesting, but the answer that the authors propose need more direct evidence. Here are some suggestions and comments.

1. From the NMR (Fig 4C), T1 footprinting (Fig 5B) and SHAPE data (Fig S5), the authors suggest a mechanism in which: (a) hnRNP A1 binding to the pri-18a terminal loop leads to partial opening/melting; and (b) destabilization of the pri-18a stem region. (a) The footprinting and SHAPE data show changes in signal when hnRNP A1 is added. However, there is less evidence for the statement "leads to partial opening/melting terminal loop" in the data presented. (b) The authors explain that there is stem destabilization hypothesis using data from NMR (Fig 4C), footprinting (5B) and SHAPE (data not shown? - Line 393- I thought this was not acceptable.). The current SHAPE data in Fig S5 does not clearly support the destabilization model since there is reduced SHAPE reactivity in the stem region when UP1 is present.

In our revised version, we added SHAPE data that provide additional mapping of the UP1 binding site covering both UAG motifs in the apical loop. As the reviewer correctly notes there is reduced SHAPE reactivity. However, this reduced reactivity does not map to the stem region, but covers the binding site of hnRNP A1, and directly flanking residues in the apical loop. This is expected to result in reduced SHAPE reactivity, and is therefore fully consistent with the binding of both RRM domains to the apical loop. We have updated **Fig. 5d** for better indicating the binding site and nucleotides that interact with hnRNP A1.

The presence of base pair dynamics and changes thereof induced by A1 binding is demonstrated by our NMR data with the pri-mir-18a and the 17-mer RNAs that represents the upper stem-apical loop region (Figure 4c,d,e). First, the fact that imino protons are not observable for the upper stem-apical loop region in both the pri-mir-18a 71-mer and for the 17-mer RNA (Figure 4c,d) shows that this region is dynamic - even though base pairs are formed. The presence of these base pairs (although dynamic), is unambiguously demonstrated by the presence of J-couplings across the hydrogen bonds seen in the (Py) H(CC)NN-COSY experiment with the wild type 17-mer (Figure 4e).

Second, upon A1 binding a number of imino correlations in the central stem region of the pri-mir-18a 71-mer become severely broadened and are not detectable anymore, while imino signals in the lower stem region remain detectable (Figure 4c, we have added green arrows to make this clearer). Given that A1 binds to the terminal loop, this is consistent with an allosteric destabilization that renders the stem base pairs more dynamic and broadens the imino signals. Note, that the footprinting experiments also show an increased reactivity to enzymatic probing upon hnRNP A1 binding to the terminal loop (Figure 5b, indicated by green arrows).

We mentioned in the revised manuscript and the point-by-point response that SHAPE is not sensitive enough to pick up the destabilization of the central stem part (beyond the A1 binding site in the terminal loop). While we observed increased SHAPE reactivity for base pairs in the stem region upon A1 binding, these are not clearly visible anymore when averaging the three replicates of the SHAPE experiment and are thus not statistically significant. The lack of change in SHAPE reactivity in the stem region beyond the apical loop is consistent with the suggestion that A1 binding to the terminal loop does not disrupt base-pairing in the flanking stem region but merely renders the base pairs more dynamic, while still being stacked.

2. The resolution from NMR is not sufficient to derive high-resolution spatial information. And both footprinting and SHAPE data would benefit from having controls that do not completely change the RNA structure, such as mutant hnRNP A1. This was the original points #5 and #3. It is a simple question about whether a protein mutation in RRMs lead to a change in the footprint or SHAPE or NMR. Moreover, whether a protein mutation in RRMs lead to changes in processing. If the primary novelty that the authors are suggesting is the way UP1 sits on the terminal loop, this is a rather simple experiment and much more telling than elaborate chimeras where the RNA structures

have changed a lot. The reviewer suggests that the authors look for changes in RNA when one of the RRM is mutated. Since you know the complex structure, mutant proteins would be better as controls than in the absence of protein.

We respectfully disagree. NMR chemical shifts and NOE correlations are very sensitive to structure and environmental changes. The changes in NMR correlation spectra map the binding site with the RNA and fully agree with the RNA binding surface seen in the crystal structure. Furthermore, the 1:1 model is confirmed by NMR PRE and RDC data and unambiguously demonstrated by our ITC, static light scattering and SAXS data. We already explained that we could not find conditions for a full NOE based structure determination.

As for using **hnRNP A1 mutants**, it is not clear from the reviewer's comment, which additional mutants he/she is suggesting that we should use. Mutating the RRMs in hnRNP A1/UP1 leads to weaker or no binding to the RNA as we have already shown with the affinities for individual RRM1 and RRM2 alone by EMSA (**Suppl. Figure 1c**), NMR (**Figure 2**) and ITC (**Figure 2; Suppl. Fig. 2**). Moreover, we already report that mutations of residues in individual or both RRMs in nuclear localized UP1 (which represents the effects of full-length hnRNP A1) impair pri-mir-18a processing (**Suppl Fig 1a and Suppl Table 1**). These mutations are:

UP1-M9-FD2: F148D/F150D

UP1-M9-FD12: F57D/F59D/F148D/F150D

UP1-M9-FD12a: F17D/F57D/F59D/F108D/F148D/F150D and

UP1-M9-FA12b: F17A/F57A/F59A/F108A/F148A/F150A).

Perhaps the reviewer has overlooked the data? In fact, all of these mutations were performed within the context of UP1-M9, which also lacks the glycine-rich intrinsically disordered C-terminal tail of hnRNP A1. This demonstrate that the glycine-rich region is not required for the processing activity of the protein, while RNA binding mutations in RRMs are not functional, fully consistent and clearly supporting our model.

We have ensured to explicitly referring to these data in the further revised text. The lack of pri-mir-18a processing is fully consistent with the requirement of RNA binding by both domains, for which we have provided multiple experimental data, i.e. EMSA, NMR and ITC. In conclusion, we simply do not see that performing further mutational analysis will provide any additional support or controls for the model that we propose.

3. The 16/18 chimera experiments are rather indirect. As the authors showed, various RNA elements can have effects on processing, and why certain mutations they tested affect Drosha processing is still unclear. And it is unclear how one feature (such as the 5GC clamp) would interact with another. For example, how the pri-mir-18a 5GC interacts with hnRNP A1 might be different from the wild type miR-18a. Instead of controlling with miR16, it would be better to control with hnRNP A1 mutations.

Importantly, we have already demonstrated that the affinity of hnRNP A1 to wild type pri-mir-18a and pri-mir-18a[5GC_internal] is the same (**Figure 6b**). Analyzing hnRNP A1 mutants on pri-mir-18a[5GC_internal] would provide no information as this pri-mir is not processed by Drosha in the first place (**Figure 6b**). Introduction of the pri-mir-16/18a[5GC_internal] control shows that the 5GC clamp element is tolerated by Drosha in the context of an heterologous pri-miR (Fig7). Notably, pre-mir-18a[5GC_internal] and pre-mir-16/18a[5GC_internal], generated by Drosha cleavage, are similar in size. This provides additional support to our model and confirms the requirement of structural RNA binding feature by hnRNP A1 for the specific processing of pri-mir-18a.

In future studies, it will be interesting to assess how the pri-miR variants are incorporated (or not) into the Microprocessor.

4. ITC can be used to measure both affinity and stoichiometry. Can the authors use this instead of just the affinity values to determine the ratios?

It is unclear to us what the reviewer refers to, and what he/she means with "determine the ratios". We fully agree that an advantage of ITC experiments – beyond being a label-free solution technique – is that

they provide information about thermodynamic parameters, affinity and stoichiometry. We indeed **report both affinities (K_D s) and stoichiometries in Table 1** and discuss these data in the text.

5. Line 495 “How does binding of a regulator to the terminal loop of pri-miRNA affect Drosha...”: Current model of primary microRNA processing includes that most of the terminal loop is critical for Drosha processing, even before any loop binding proteins are present. And it also suggests that the RNA motifs/structure are linked, thus making it difficult to assume that certain features are additive. Please refer to Nguyen et al. 2015, Kwon et al. 2015, Auyeong et al. 2013, Partin et al. 2017, and Roden et al., 2017. What the authors propose—that hnRNPA1 melts the loop and somehow propagates the changes to the Drosha cut site—is intriguing but requires more evidence. For example, the direct effect of hnRNP A1 needs to be shown, via reconstitution or using appropriate protein mutants. This is especially important since various structural and biochemical evidence suggest that a lot of the stemloop is already contacting Drosha and DGCR8. Moreover, the manuscript would be stronger with a discussion that can reconcile the differences in the models.

Thank you for this comment, indeed we agree that it will be interesting to present a more comprehensive discussion of our findings with respect to current views of the molecular mechanism involving pri-miR processing by the Microprocessor.

With respect to protein mutations, please refer to our explanations above (points 2 and 3): We do not think that any additional analysis of **hnRNP A1 mutants**, i.e. beyond those that we have already included in our study (**Supplementary Fig. 1, Supplementary Table 1**) will provide any relevant further insight. We also have already reasoned that **reconstitution experiments** with the Microprocessor are beyond the scope of the current manuscript, but are certainly an avenue to pursue in future studies.

With respect to **discussing the role of hnRNP A1 in enhancing pri-mir-18a processing**, we have revised and expanded the discussion. We now present a more clearly structured discussion and propose three possible mechanisms of action of hnRNP A1 in enhancing the processing of pri-mir-18a, which are not mutually exclusive. hnRNP A1 binding may 1) promote Drosha processing by stem destabilization and thereby enhancing the recruitment and/or assembly of an active Microprocessor substrate complex, 2) enhance the assembly of the correct orientation of pri-mir-18a onto the Microprocessor by generating a thermodynamic or conformational asymmetry in the stem, and/or 3) increase the accessibility of the Microprocessor to the pri-mir-18a hairpin in the context of the complete oncomir-1 cluster.

To our knowledge, including from the papers suggested by this referee, it is not clear if there is a specific order of loop binding by cofactors or the microprocessor. In fact, there are still ongoing discussions and controversial views about the role of certain sequence and or structural features in pri-miRNAs and their role for processing by the Microprocessor. In this respect, we are convinced that our work provides an important contribution for highlighting the role of terminal loop binding by RNA binding proteins, where molecular and structural mechanisms are largely unknown.

In further expanding the discussion, we relate our findings into context with existing models about Microprocessor binding and its regulation by cofactors, such as RNA binding proteins or heme. In fact, Partin et al (Nature Commun 2017) propose that heme binding by DGCR8 can be required to ensure the correct orientation of Drosha on pri-miRNA stems. A similar role may extend to terminal loop binding RBPs such as A1, as we suggest as one possible function.

We believe that we have addressed the useful suggestion by the reviewer to provide a more extensive discussion. The significantly revised discussion presents 1) hypotheses for possible functions of hnRNP A1 binding and 2) an expanded discussion of existing models of Microprocessor mechanisms, thereby referring to published data and the manuscripts as suggested by the referee.

Reviewer #2 (Remarks to the Author):

I would urge the authors to show the EMSA results as I recommended in the previous review for the following reasons:

1. *The interpretation of ITC results depends on type of models one uses to fit the data, where the EMSA gives a straightforward binding profile without any assumption.*

We agree that EMSA and ITC provide different and complementary means of studying binding. However, given that we work with purified components, in our view ITC is preferable as it is more precise and accurate to assess the differences of binding affinities with the different RNAs. Moreover, ITC is a pure solution method and is not affected by interactions with the gel. We show EMSA data with wildtype and mutant proteins in **Fig. 1** and **Suppl. Figure 1**.

2. *ITC does not always work, as authors have shown in UP1+17mer titration, even though NMR appears to indicate it should work and give an expected result.*

We agree. In fact, we have already shown the NMR titrations with the 17-mer (wildtype and A35C variant) in **Suppl. Fig 3c and d**.

Please note, that the unusual ITC curve observed for the wildtype 17-mer is consistent with the presence of two events to produce enthalpy change, namely an unfolding of the 17-mer hairpin coupled with protein binding, and thus supports our model.

The A35C mutant 17-mer exhibits only one single-stranded UAG motif for binding to UP1 (as the second motif is part of a highly stable RNA helix and thus not accessible). Consistently, the A35C shows an expected ITC binding curve with a 1:1 stoichiometry and micromolar affinity (Supplementary Figure 2d), comparable to the 7-mer RNA with a single UAG motif. The fact that the wildtype pri-mir-18a exhibits one UAG motif in the terminal loop and a second one in the weak and dynamic upper stem region suggests that binding of the UP1 tandem domains requires melting of the stem region flanking the terminal loop. This is consistent with the ITC binding curve observed for the wildtype 17-mer RNA (Supplementary Figure 2d), which shows unusual concentration-dependent enthalpy changes that cannot be fitted to a simple binding event, and rather may reflect coupled RNA unfolding and binding of the protein. That a similar complex titration curve is not observed for the 71-mer pri-mir-18a RNA is consistent with the fact that this RNA is not unfolded completely, as hnRNP A1 binding leads merely to local melting of the terminal loop and dynamics of the flanking stem region.

I understand the EMSA gel pictures may look messy. But it will give readers a complete realistic and unhinged view about the complexes: whether they are in fast, slow or intermediate exchange regimes in addition to NMR results, which are usually measured at much higher concentration.

For quantitative analysis and assessment of structural features we have used NMR and ITC to study variations in RNA ligands, while the overall evidence for UP1 and RRM binding is demonstrated by EMSAs (**Fig. 1**, **Suppl. Fig. 1**).

I am otherwise satisfied with all other responses from the authors

Thank you.